# Soil moisture–atmosphere feedback dominates land carbon uptake variability

Vincent Humphrey[1✉], Alexis Berg[2], Philippe Ciais[3], Pierre Gentine[4], Martin Jung[5], Markus Reichstein[5], Sonia I. Seneviratne[6] & Christian Frankenberg[1,7]

Year-to-year changes in carbon uptake by terrestrial ecosystems have an essential role in determining atmospheric carbon dioxide concentrations[1]. It remains uncertain to what extent temperature and water availability can explain these variations at the global scale[2–5]. Here we use factorial climate model simulations[6] and show that variability in soil moisture drives 90 per cent of the inter-annual variability in global land carbon uptake, mainly through its impact on photosynthesis. We find that most of this ecosystem response occurs indirectly as soil moisture–atmosphere feedback amplifies temperature and humidity anomalies and enhances the direct effects of soil water stress. The strength of this feedback mechanism explains why coupled climate models indicate that soil moisture has a dominant role[4], which is not readily apparent from land surface model simulations and observational analyses[2,5]. These findings highlight the need to account for feedback between soil and atmospheric dryness when estimating the response of the carbon cycle to climatic change globally[5,7], as well as when conducting field-scale investigations of the response of the ecosystem to droughts[8,9]. Our results show that most of the global variability in modelled land carbon uptake is driven by temperature and vapour pressure deficit effects that are controlled by soil moisture.

Improving the ability of Earth system models (ESMs) to correctly reproduce the observed variability in land carbon fluxes is essential for building confidence in projections of the long-term response of the carbon cycle to a warming and changing climate[10]. This research agenda has been evolving rapidly in the past decade thanks to coordinated model comparison experiments[11,12], theoretical advances[13], model developments[14,15], as well as new observations from ground-based networks[16,17] and satellite platforms[18]. Yet, the spread among ESMs remains substantial[19,20] and highlights the need to better constrain the sensitivity of increasingly complex biogeochemical models to changes in atmospheric and hydrological drivers such as radiation[21], temperature[7], soil water availability[3] and vapour pressure deficit (VPD; a measure of atmospheric dryness that depends on air temperature and humidity). In particular, it remains unclear whether temperature or soil moisture is the dominant driver of the inter-annual variability (IAV) in land carbon uptake at the global scale[2–5]. Here, we investigate the extent to which temperature, VPD and soil moisture effects co-vary as a result of soil moisture–atmosphere feedback, and reconcile conflicting assessments of the sensitivity of global carbon fluxes to these variables.

Soil moisture drought is one of the key prerequisites for the development of extremely high temperatures[22–24], whereas atmospheric dynamics control the onset of such extremes[25]. During droughts, low soil moisture content limits evapotranspiration, which is the most efficient surface cooling flux[26]. This modification of the surface energy balance increases the air temperature, lowers the relative humidity and

thus raises VPD. The importance of such soil moisture–atmosphere feedback, hereafter referred to as land–atmosphere coupling (LAC), is confirmed by both models and observations[27–29]. In current carbon cycle models, the impacts of soil moisture, temperature and VPD on ecosystem productivity and respiration are usually parameterized using stress functions. Typically, simulated photosynthesis rates are limited by low soil moisture content and extreme temperatures via a scaling of $V_{cmax}$ (the maximum rate of Rubisco carboxylase activity)[30] or through a downregulation of stomatal conductance ($g_s$) in response to VPD, relative humidity or a soil water stress function[31,32]. Ecosystem respiration and fire occurrences are also controlled by soil moisture content, temperature or atmospheric dryness[33,34]. Because of this situation, the overall influence of soil moisture can potentially occur as (1) a direct impact on photosynthesis and respiration processes through the soil water stress regulation or (2) as an indirect response to extreme temperature and VPD anomalies resulting from LAC.

Here, we investigate the magnitude of these two different causal pathways (that is, direct and indirect) using coupled climate model simulations from the Global Land-Atmosphere Coupling Experiment, Coupled Model Intercomparison Project 5 (GLACE-CMIP5)[6] (Methods). To identify the overall influence of soil moisture variability on carbon fluxes and atmospheric conditions, we use an experiment (experiment A) in which the non-seasonal variability in soil moisture is artificially removed. This is achieved by forcing the soil moisture in experiment A to follow the mean seasonal soil moisture cycle calculated from a reference

[1]Division of Geological and Planetary Sciences, California Institute of Technology, Pasadena, CA, USA. [2]Department of Earth and Planetary Sciences, Harvard University, Cambridge, MA, USA. [3]Laboratoire des Sciences du Climat et de l'Environnement, CEA CNRS UVSQ, Gif-sur-Yvette, France. [4]Department of Earth and Environmental Engineering, Columbia University, New York, NY, USA. [5]Department of Biogeochemical Integration, Max Planck Institute for Biogeochemistry, Jena, Germany. [6]Institute for Atmospheric and Climate Science, ETH Zurich, Zurich, Switzerland. [7]Jet Propulsion Laboratory, California Institute of Technology, Pasadena, CA, USA. ✉e-mail: vincent.humphrey@caltech.edu

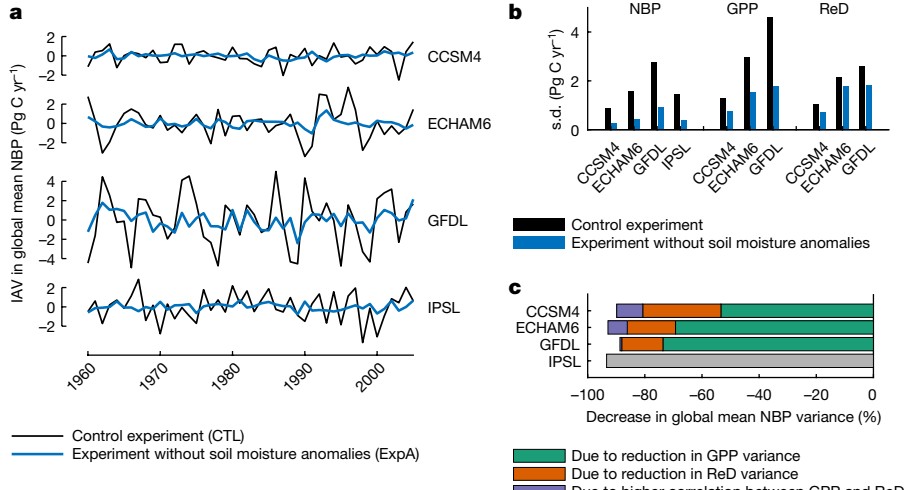

**Fig. 1 | Carbon fluxes in CTL and experiment A. a**, IAV in global mean NBP (mean-centred and de-trended) as simulated by four ESMs (CCSM4, ECHAM6, GFDL and IPSL) in coupled model experiments with (CTL) and without (experiment A; ExpA) anomalies in soil moisture. Positive NBP indicates carbon uptake. **b**, Standard deviations of global mean NBP, GPP and respiration and disturbance (ReD) in the two experiments. **c**, Drivers of change in global mean NBP variance (Supplementary Information section 1). Global mean NBP variance decreases in the experiment with prescribed seasonal soil moisture mainly because GPP variance is reduced. GPP and ReD fluxes are not available for the IPSL model.

control simulation (CTL) (Extended Data Figs. 1, 2). Experiment A thus simulates the temperature, VPD and carbon fluxes that would occur under climatologically normal soil moisture conditions. We note that sea surface temperatures (SSTs) are identical in experiments A and CTL. This ensures that the main differences between experiment A and CTL are due to the different soil moisture conditions and are not caused by differences in SST patterns (Methods). Using this framework, previous studies have shown that suppressing the non-seasonal soil moisture variability in experiment A strongly reduces the magnitude of temperature and VPD extremes compared to the control simulation[6,27,35] (Extended Data Fig. 3). Here, by comparing the carbon flux anomalies of experiment A with those of the control simulation, we are able to estimate the overall magnitude of soil moisture effects (that is, direct and indirect effects) on the IAV of net biome production (NBP; which represents the net land carbon uptake). Because we focus on the IAV, all presented figures are based on anomalies (de-seasoned and de-trended data) from the period 1960–2005, unless otherwise noted.

Our results show that suppressing non-seasonal variability in soil moisture leads to a 91% (standard deviation of ±2.3%) decrease in the variance of global mean NBP, consistently across all of the four participating climate models (Fig. 1a, Supplementary Table 1). In other words, without soil moisture variability, the IAV of net land carbon uptake is almost eliminated. This primarily occurs because of a reduction in the IAV of gross primary production (GPP) (Fig. 1b, c, Supplementary Table 1) and to a lesser extent because of a reduction in the IAV of ecosystem respiration and disturbance fluxes (the sum of autotrophic and heterotrophic respiration, fires and any other modelled disturbance). As explained above, both direct soil moisture effects and indirect temperature and VPD effects related to LAC can be responsible for the widespread reduction of NBP variability occurring in experiment A (Fig. 2a).

Using a sensitivity analysis (equations (1), (2), Supplementary Figs. 1–3) of the local model response to anomalies in soil moisture, temperature, VPD and shortwave solar radiation in CTL versus experiment A, we isolate the contributions of direct soil moisture effects (Fig. 2b) versus indirect effects (Fig. 2c) to the overall reduction in NBP variability (Fig. 2a). Regionally, direct soil moisture effects are found in both temperate and tropical biomes, whereas indirect effects occurring through the feedback on temperature and VPD are mostly concentrated in semi-arid and tropical regions. Our sensitivity analysis also shows

that most of the reduction in NBP variability found in experiment A occurs because of a reduction in the variance of the climatological drivers, rather than because of a change in the sensitivity of NBP to these drivers (Extended Data Fig. 4). These findings demonstrate that soil moisture can affect carbon uptake variability in two different and equally important ways. First, soil moisture variability has direct effects on NBP, mostly because plant photosynthesis is reduced when soils become dry below a certain threshold (Fig. 2b); second, it enhances temperature and VPD anomalies through LAC, thus leading to indirect effects on NBP (Fig. 2c, Extended Data Fig. 5). Importantly, some regions can be more sensitive to indirect effects (that is, soil moisture feedback mechanisms on temperature and VPD) than to direct soil moisture effects (Extended Data Fig. 6). We note that because disentangling the individual contributions of temperature and VPD to NBP variability is not straightforward, only their joint contribution is reported here (see Methods for a discussion).

When aggregating these results to the global scale (Fig. 3a), we find that indirect effects alone are on average (across models) responsible for most (60%) of the global NBP IAV, whereas direct soil moisture effects account for only 20%. Suppressing direct and indirect effects together leads to a net decrease in NBP variance of about 90% (consistent with Fig. 1) as a result of the positive covariance between the direct and indirect effects (Supplementary Tables 2, 3). Finally, the temperature (T) and VPD effects that are independent of soil moisture conditions and still persist in experiment A (NBP$_{nonLAC}^{T\&VPD}$) account for only 9% of the overall global NBP variability, whereas radiation effects account for the remaining 11%. As a result of spatial aggregation (Fig. 3b), indirect effects also tend to increase in relative importance as they are spatially more coherent (probably owing to atmospheric mixing) and do not average out as fast as the direct effects[2]. In summary, the largest fraction of the global mean NBP IAV is driven by anomalies in temperature and VPD that represent an indirect response to soil moisture variability (given that they do not occur in its absence, as demonstrated by the experiment). This finding reconciles opposing perspectives on the roles of temperature versus water availability[2–5], because the apparent importance of either driver actually depends on whether the indirect (feedback) effects are attributed to temperature or soil moisture (see Extended Data Fig. 7, Supplementary Fig. 5). Although it is not possible to replicate the factorial experiment with observations (this would

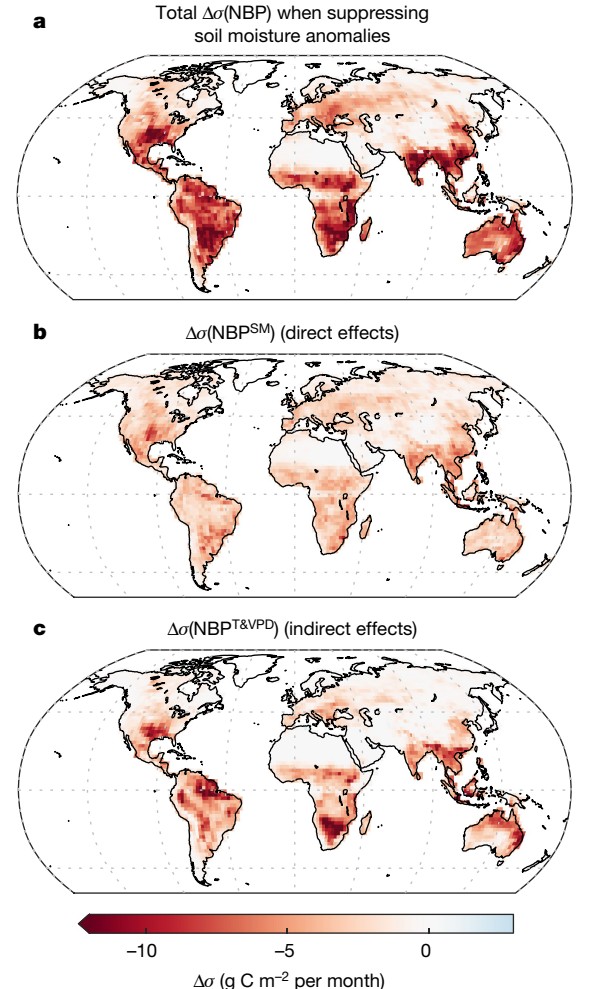

**a** Total $\Delta\sigma$(NBP) when suppressing soil moisture anomalies

**b** $\Delta\sigma$(NBP$^{SM}$) (direct effects)

**c** $\Delta\sigma$(NBP$^{T\&VPD}$) (indirect effects)

$\Delta\sigma$ (g C m$^{-2}$ per month)

**Fig. 2 | Direct and indirect soil moisture effects on NBP variability.**
**a**, Change in annual NBP standard deviation ($\Delta\sigma$) when prescribing seasonal soil moisture. **b**, Change caused by a direct response to the suppressed soil moisture (SM) variability. **c**, Change caused by the reduced variability of temperature (T) and VPD (that is, the indirect effects of suppressing soil moisture variability). Negative values in **a**–**c** indicate a decrease of the variability in experiment A compared to CTL. The median across the four models is shown.

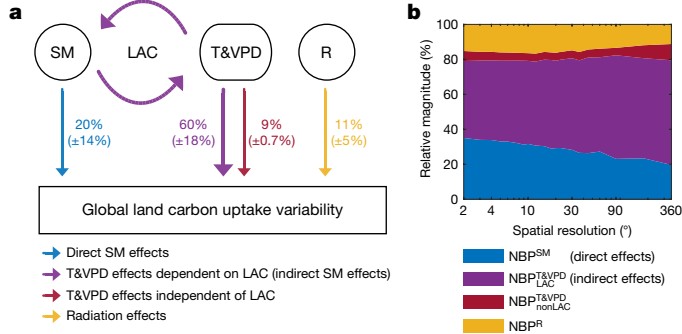

**Fig. 3 | Drivers of NBP IAV. a, b**, Contribution of meteorological drivers to NBP IAV: direct soil moisture effects (NBP$^{SM}$), indirect LAC-dependent temperature and VPD effects (NBP$^{T\&VPD}_{LAC}$), non-LAC-dependent temperature and VPD effects (NBP$^{T\&VPD}_{nonLAC}$) and radiation (R) effects (NBP$^{R}$) globally (**a**; mean of the four models ±1$\sigma$) and from local to global scales (**b**).

These results also improve our understanding of the sensitivity of land carbon uptake IAV to tropical mean temperature[40,41], which has been used to constrain coupled climate model projections[7,42]. Here, we find that the IAV of mean tropical land temperature is barely changed in the experiment with prescribed soil moisture (Extended Data Fig. 10). This is because suppressing soil moisture anomalies reduces temperature extremes only in a couple of hotspot regions (Fig. 4b, Extended Data Fig. 3) with little impact on the overall tropical mean. Thus, although the IAV in global land carbon uptake has been empirically found to be sensitive to tropical mean temperature in numerous studies[5,41], our results suggest that this sensitivity does not represent a strong mechanistic link, and thus might not necessarily represent the most adequate model constraint. In fact, the El Niño Southern Oscillation and SST in general may be the confounding driver of both tropical mean temperature and the precipitation patterns that cause the soil moisture anomalies leading to NBP variability.

In conclusion, we show that the IAV in land carbon uptake simulated by ESMs is primarily driven by anomalies in temperature and VPD, which are themselves controlled by soil moisture variability. These indirect soil moisture effects occur through LAC and account for 60% (±18%) of the simulated global land carbon uptake IAV. They explain why the simulated global NBP variability (1) mainly arises from tropical and semi-arid regions[37,38], which are known hotspots of LAC[6,36,43], (2) is predominantly a temperature and VPD response (at the global scale) according to land surface models and empirical sensitivity analyses[2,5] and (3) is also largely dependent on soil moisture variability according to coupled climate simulations[4]. Our results reveal that soil moisture–atmosphere feedback mechanisms represent a dominant source of variability in global carbon uptake and thus reconcile previous conflicting assessments[2–5]. To some extent, we note that these findings might be symptomatic of how land surface models were developed in the first place. Parameterizing a strong sensitivity of carbon uptake to observed VPD or temperature can constitute a simpler way for a land-surface model to achieve good skill, especially when soil water stress and soil moisture dynamics are only represented approximately. As a result, even though models strongly agree that direct and indirect soil moisture effects together dominate land carbon uptake variability, the actual partitioning between direct and indirect effects may be more dependent on modelling approaches. More generally, our results illustrate the importance of differentiating estimates of ecosystem sensitivity to natural droughts, as opposed to artificial droughts (for example, rainfall exclusion experiments), given that only the former incorporates LAC and its impact on temperature and humidity. Because soil and atmospheric dryness do not equally respond to climate change[27,44], the direct and indirect soil moisture effects identified here might affect future NBP in different ways. Since current climate models

require manipulating soil moisture everywhere on the planet), we assess the degree to which the reference simulations reflect real observations. Evaluating the control simulations against observational estimates, we find that the modelled sensitivity of global NBP IAV to the different meteorological drivers (Fig. 3) agrees well with two independent observational products (Extended Data Fig. 8). Taking into account the uncertainty of these observations, the spatial patterns of NBP IAV simulated by the models are also in reasonable agreement with real-world variability (Supplementary Fig. 6; see discussion in Methods).

More generally, our results show that the areas where NBP IAV is the largest overall (Fig. 4a) often correspond to those where the reduction of temperature and VPD variability due to prescribing soil moisture is the strongest (Fig. 4b, c). In other words, NBP variability tends to be larger where LAC is stronger (Fig. 4d). These known hotspots of LAC[36] match well with earlier studies that suggested that semi-arid regions dominate global NBP IAV[37,38], even though our analysis refines these previous findings (Extended Data Fig. 9) by also including regions that are usually classified as temperate or humid but that are affected by LAC for only a few dry months during the year (for example, eastern Europe[22], Amazon basin[39]).

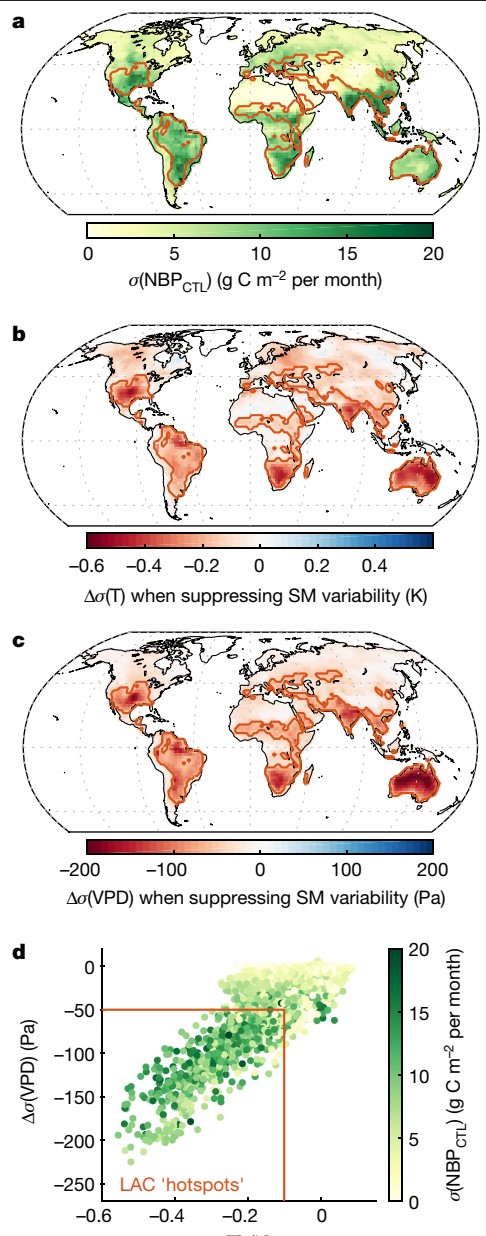

**Fig. 4 | NBP variability and LAC hotspots. a**, Simulated NBP IAV in the control simulation (median across the four models). **b**, **c**, Median change in the standard deviation of temperature (**b**) and VPD (**c**) when suppressing non-seasonal soil moisture variability (standard deviation in experiment A minus standard deviation in CTL). **d**, Combined representation of all the grid points in **a**–**c**. The overall IAV of NBP (colour scale) tends to be higher in regions that have a strong LAC effect. For visualization purposes, arbitrary thresholds in **d** are used to highlight hotspots of LAC in **a**–**c**.

have a large spread in their representation of vegetation response to dryness[45] and of LAC strength[46], this could introduce uncertainties in the feedback that are difficult to diagnose from offline land surface model evaluation efforts[47], with potentially large impacts on carbon fluxes, as demonstrated here. We also note that long-term changes in vegetation structure and composition might alter the ecosystem's future response[4] to and control[9,48,49] of soil moisture–atmosphere feedback. Thus, more physical and holistic representations of the response of vegetation to soil and atmospheric dryness might have a strong potential to reduce key uncertainties in current projections of future terrestrial carbon fluxes.

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

## Methods

### Model experiment

The presented results are based on the Global Land–Atmosphere Coupling Experiment – Coupled Model Intercomparison Project phase 5 (GLACE-CMIP5) numerical experiment[6]. This model experiment was originally designed to investigate soil moisture–climate feedback mechanisms under historical and future scenarios, and notably their impact on extreme heat events[6]. Its experimental design is inspired from the original GLACE experiment[43], which focused on the role of soil moisture in seasonal weather predictability. Six ESMs were used for global climate simulations: the Community Climate System Model 4 (CCSM4), the European community ESM (EC-Earth), the European Centre/Hamburg Model 6 (ECHAM6), the Geophysical Fluid Dynamics Laboratory model (GFDL), the Institut Pierre-Simon Laplace model (IPSL) and the Australian Community Climate and Earth System Simulator (ACCESS). Model outputs for carbon fluxes are available only for four models (CCSM4, ECHAM6, GFDL and IPSL), and the availability of certain variables is limited in some cases (Supplementary Table 4), which explains why some analyses cannot be conducted with all models (for example, Fig. 1c).

The control (CTL) and soil moisture (experiment A) experiments consist of coupled atmosphere–land simulations (Extended Data Fig. 2) using prescribed SSTs, sea ice, land use and atmospheric $CO_2$ concentrations from each of the model's fully coupled reference CMIP5 runs (except for CCSM4, where the reference CMIP5 run itself is used as the control simulation). Unlike so-called 'offline' simulations, in which a land surface model is driven by a fixed meteorological forcing, a coupled simulation resolves water and energy exchanges between the land and the atmosphere, allowing land processes to feed back to the atmosphere and influence it locally. The model simulations cover the historical period since 1950 and the 21st century (RCP8.5 scenario). Further details documenting the control experiment, including the description of the atmospheric and land model components, can be found in Seneviratne et al.[6]. The only forced difference between the CTL and experiment A simulations is the soil moisture variability. In experiment A, soil moisture is prescribed to a reference climatology (seasonal cycle) calculated from the control run over the period 1971–2000 (Extended Data Fig. 1). Thus, the main difference (on a climatological timescale) between the two simulations is related to the change in soil moisture. It is worth noting that at finer, meteorological, timescales (for example, daily time series), the internal variability inherent to general circulation models will also lead to differences between the two simulations.

Prescribing soil moisture implies that the water balance is not necessarily conserved. An investigation of this imbalance with the Community Earth System Model (CESM)[50] showed a positive net imbalance (that is, the sum of all water additions and subtractions) of the order of +8% globally (relative to the annual mean precipitation), associated with an overall increase in land evapotranspiration. We note that in some specific regions, less water may be added than is removed (negative imbalance) and that temperature extremes are found to be reduced in both cases (positive or negative imbalance) as a result of the suppressed LAC. Although there is no apparent impact on global mean precipitation[50], there are some changes in the distribution of precipitation (for example, an increase in extreme events over the tropics[51]). We do not expect changes in precipitation between CTL and experiment A to have any impact on carbon fluxes (because soil moisture is prescribed).

To enable a consistent comparison, we re-grid all model outputs to a common resolution of 2° using conservative re-gridding and compute monthly averages. The entire analysis presented here is focused on the IAV over the period 1960–2005. We note that VPD is first calculated from daily averages of temperature and relative humidity, and is only then averaged to monthly means. The IAV corresponds to the signal remaining after removing the seasonal cycle, as well as any long-term linear trend on a monthly basis (the long-term trend of each month is subtracted). For the ECHAM6 model, two grid cells located in the Tibetan plateau are discarded from the whole analysis, because spurious spikes are present in heterotrophic respiration for experiment A. We also discard Greenland and Antarctica to maintain a comparable spatial coverage for all models. Although this work focuses on anomalies (that is, deviations from the seasonal cycle), we also illustrate the seasonal cycles of NBP, GPP and respiration and disturbance simulated in CTL and experiment A in Supplementary Fig. 7. For completeness, we also provide time series of global mean soil moisture, temperature, VPD and radiation IAV (similar to Fig. 1) in Supplementary Fig. 8.

### Comparison of the control simulations with observational estimates

We evaluate the simulated IAV of NBP, soil moisture, temperature and VPD against available observations in Supplementary Figs. 6, 9–11. For NBP IAV (Supplementary Fig. 6), we note that although observational estimates of NBP variability exist, they do not agree well with each other, reflecting our limited knowledge of net carbon fluxes globally[52,53] (Supplementary Fig. 6g, 'obs vs obs'). To focus on time periods in which these observational datasets are more reliable globally, we use the period 1980–2010 for the FLUXCOM RS+METEO dataset and the period 2000–2018 for the CAMS atmospheric $CO_2$ inversion. We show that models correlate with these observational estimates as much as the observations themselves correlate with each other (Supplementary Fig. 6g, 'models vs obs'). We also find that there is little consensus on the overall (de-trended) NBP IAV amplitude. The global mean NBP standard deviation of the different models ranges from 0.86 petagrams of carbon per year (Pg C yr$^{-1}$) for CCSM4 to 2.76 Pg C yr$^{-1}$ for GFDL. When comparing with observational products (Supplementary Fig. 6h), we find that— excluding FLUXCOM RS+METEO, which is known to underestimate the global NBP IAV[52]—the CAMS atmospheric $CO_2$ inversion[53] suggests a value of 0.68 Pg C yr$^{-1}$, whereas dynamic vegetation models used for the Global Carbon Project[1] suggest a range of 0.53 to 1.50 Pg C yr$^{-1}$. Thus, some models (GFDL in particular) seem to overestimate the overall NBP variability. However, regardless of how close they are to observations or other estimates, all models are unanimous that the global NBP variance is reduced by about 90% when prescribing soil moisture and that indirect effects dominate this response (Figs. 1, 3).

We evaluate spatial patterns of IAV for soil moisture, temperature and VPD against available observational datasets in Supplementary Figs. 9–11. The simulated soil moisture IAV patterns agree reasonably well with the total soil moisture from the ERA5-Land reanalysis[54] and with satellite observations of shallow soil moisture (5–10 cm depth) from the ESA CCI Combined product v4.5[55] (Supplementary Fig. 9). Regarding temperature and VPD IAV, we find that models and observational sources[56,57] are in reasonable agreement (Supplementary Figs. 10, 11). Finally, we also evaluate spatial patterns of global long-term mean GPP, which is arguably better constrained by observations than long-term mean NBP. We find that the models agree very well with the observational data[52,58] in terms of spatial patterns (Supplementary Fig. 12). For global mean GPP, two models produce a relatively high global mean GPP (of about 150 Pg C yr$^{-1}$). However, such values are not entirely unrealistic according to other satellite-based estimates (for example, Joiner et al.[59] report 140 Pg C yr$^{-1}$).

### Sensitivity analysis

In Figs. 2, 3 we reproduce the approach by Jung et al.[2], which consists of a local month-wise linear regression of the NBP model output against the main meteorological drivers (which are also deseasonalized and detrended):

$$\mathbf{NBP}^*_{s,m} = \beta^{SM}_{s,m}\mathbf{SM}_{s,m} + \beta^{T}_{s,m}\mathbf{T}_{s,m} + \beta^{VPD}_{s,m}\mathbf{VPD}_{s,m} + \beta^{R}_{s,m}\mathbf{R}_{s,m}, \qquad (1)$$

where $s$ is the spatial index (grid point), $m$ is the month index (1 to 12) and $\beta$ are regression coefficients. **NBP**, **SM**, **T**, **VPD** and **R** are $N \times 1$ vectors,

where $N$ is the number of years; **NBP** denotes the net biome production anomaly, **SM** represents the total soil moisture anomaly, **T** denotes the 2-m air temperature anomaly, **VPD** represents the vapour pressure deficit anomaly and **R** is the surface downward solar radiation anomaly. In the main text, the four components of equation (1) are referred to using the more compact notation:

$$NBP^* = NBP^{SM} + NBP^T + NBP^{VPD} + NBP^R, \qquad (2)$$

where $NBP^{SM}$, $NBP^T$, $NBP^{VPD}$ and $NBP^R$ correspond to the soil-moisture-driven, temperature-driven, VPD-driven and radiation-driven NBP, respectively, and $NBP^*$ is the overall result of the regression. This regression is applied to CTL and experiment A simulations separately (each regression is referred to using the appropriate notation $NBP^*_{CTL}$ or $NBP^*_{ExpA}$). In Fig. 2b, c, the difference in annual NBP variability is calculated by subtracting the standard deviation of the components of equation (2) from both experiments (for example, $\Delta\sigma(NBP^{SM}) = \sigma(NBP^{SM}_{ExpA}) - \sigma(NBP^{SM}_{CTL})$).

Because this statistical approach does not incorporate other potential sources of NBP variability as explanatory variables (ecosystem memory in particular, but also fires) and can only capture linear relationships within a given month, it should not be expected to capture the full complexity of ESM outputs. Our evaluation shows that this approach is able to reproduce a correct NBP IAV at the global (Supplementary Figs. 1, 2) and local (Supplementary Fig. 3) scales, although the overall NBP variability is generally underestimated because of the reasons mentioned above. We also apply this statistical approach to two fully independent observational estimates of NBP fluxes. We use the FLUXCOM RS+METEO dataset (GSWP3 version) over the period 1981–2010[52], which is a machine learning-based upscaling of flux tower measurements, and the CAMS v18r3 dataset[53], which is an atmospheric $CO_2$ inversion, over the period 2000–2018. We find that the overall partitioning of global NBP IAV between the different drivers is similar to what models are suggesting (Extended Data Fig. 8). The ability of the regression to reproduce these observational estimates is shown in Supplementary Fig. 13. For FLUXCOM, the sensitivity analysis is able to capture the variability almost perfectly. This is only possible because we use the same predictors as the ones used by the machine learning algorithms (that is, the GSWP3 meteorological forcing[60]). As a result, there is perfect internal consistency between FLUXCOM NEE and its predictors. For the CAMS inversion, however, such internal consistency does not exist. Using ERA5-Land[54] soil moisture, temperature, VPD and radiation as predictors, we find that the sensitivity analysis agrees relatively well with the models, even though it underestimates the magnitude of CAMS NBP anomalies at the global scale. Locally, this regression performs moderately well (Supplementary Fig. 13f), which is nonetheless a reasonable result when considering the very high uncertainty of regional NBP anomalies when they are derived from $CO_2$ inversions at the sub-continental scale[53].

Of particular interest to this work is the difference in NBP variance between CTL and experiment A (Fig. 2a). We find that this difference can be reproduced very well by the sensitivity analysis for three out of the four models (Supplementary Fig. 4). Differences are underestimated for the CCSM4 model, but this seems to occur rather uniformly and most spatial patterns are preserved (thus the ratio of NBP variance between CTL and experiment A estimated from the regression is close to the actual one; see Supplementary Table 3). Closer inspection of the regression residuals suggests that ecosystem memory and lag effects (which cannot be captured by equation (1)) might be particularly important for this model. It is interesting to note that for some models (for example, GFDL), the NBP variance can also increase locally when seasonal soil moisture is prescribed (Supplementary Fig. 4). This occurs only in a few arid regions that have almost no NBP variability in the control simulation and where soil moisture is extremely low except during occasional wet years. Prescribing a mean seasonal soil moisture in those regions causes small amounts of soil water to be available every year (instead of every few years), which increases the overall NBP variability.

Finally, several alternative formulations to equation (1) were tested. The chosen formulation (equation (1)) is the one that best reproduces the model NBP outputs. Potential alternative formulations may consist in (i) using only soil moisture, temperature and radiation, as in Jung et al.[2]; (ii) including an interaction term between temperature and soil moisture instead of VPD; (iii) replacing VPD by relative humidity. Using any of these three alternative formulations does not affect the main finding of the study, that is, that most of the global NBP variability is driven by indirect soil moisture effects (see Supplementary Figs. 5, 14, 15).

## Joint analysis of temperature and VPD effects

In Figs. 2, 3 the contributions of temperature and VPD are represented as a sum ($NBP^{T\&VPD} = NBP^T + NBP^{VPD}$). This is because temperature and VPD are correlated to some extent (VPD is calculated from the temperature and the relative humidity), so that the ability of the sensitivity analysis to attribute NBP anomalies to either one of these two variables (that is, temperature versus VPD) might be limited in some cases. We recognize this potential limitation by analysing the joint contribution of these two variables. For completeness, individual contributions are also illustrated in Extended Data Figs. 4, 5. With the caveats mentioned above, Extended Data Fig. 4 shows that VPD has a much larger role than temperature in the reduction of NBP variability occurring between CTL and experiment A. However, this does not mean that temperature is less sensitive than VPD to prescribing soil moisture. Rather, Extended Data Fig. 5 shows that the sensitivity analysis attributes more NBP variability to VPD to begin with, but that both the VPD-driven and temperature-driven NBP variability are reduced in experiment A.

## Variance contributions at different levels of aggregation

In Fig. 3, Extended Data Fig. 7 and Supplementary Figs. 5, 14–16, the contribution of different drivers to $NBP_{CTL}$ variance is computed at different levels of spatial aggregation. The following different levels of aggregation are used: 2°, 3°, 4°, 5°, 6°, 7.5°, 9°, 10°, 12°, 15°, 18°, 20°, 22.5°, 30°, 36°, 45°, 60°, 90°, 180° and 360° (that is, global). Contributions are calculated as follows. Similarly to Jung et al.[2], the different NBP time series ($NBP^{SM}$, $NBP^{T\&VPD}$ and $NBP^R$) are first aggregated to the given spatial resolution. After aggregation, the variance of the time series (that is, $\sigma^2(NBP^{SM}_{CTL})$ and so on) are computed at each grid point. Then, the variance of the T&VPD contribution $\sigma^2(NBP^{T\&VPD}_{CTL})$ is decomposed at each grid point into an LAC-dependent and non LAC-dependent contribution, as explained in Supplementary Information section 2. After that, and similar to Jung et al.[2], the global spatial average of the variances is calculated for each of the four contributions (for example, $\overline{\sigma^2(NBP^{SM}_{CTL})}$). The relative contribution of a component at a given level of spatial aggregation (as shown in Fig. 3b) is then calculated by normalizing that global spatial average against the sum of all components:

$$\text{Contribution } (NBP^{SM}) = \frac{\overline{\sigma^2(NBP^{SM}_{CTL})}}{\overline{\sigma^2(NBP^{SM}_{CTL})} + \overline{\sigma^2(NBP^{T\&VPD}_{LAC})} + \overline{\sigma^2(NBP^{T\&VPD}_{nonLAC})} + \overline{\sigma^2(NBP^R_{CTL})}}. \qquad (3)$$

Identically to Jung et al.[2], the spread in the contributions estimated by the four different models shown in Extended Data Fig. 7 is reported in two different ways. The outer uncertainty bounds represent the standard deviation of the contribution estimated by the four models. The inner uncertainty bounds represent the standard deviation between the four estimates, but after removing the mean contribution of each model across all levels of aggregation. Thus, the inner uncertainty bounds show the uncertainty in the tendency of the contribution (its change from regional to global scale).

## Data availability

GLACE-CMIP5 model outputs can be obtained from S.I.S. (sonia.seneviratne@ethz.ch). FluxCom data are available at http://www.fluxcom.org/CF-Download/. CAMS data are available from the Atmosphere Data Store at https://atmosphere.copernicus.eu/data. ERA5 and ERA5Land data are available from the Climate Data Store at https://cds.climate.copernicus.eu. VPM-GPP is available at https://doi.org/10.6084/m9.figshare.c.3789814. ESA CCI Soil Moisture is available at https://www.esa-soilmoisture-cci.org. CRU TS data are available at https://crudata.uea.ac.uk/cru/data/hrg/. GSWP3 data are available at https://doi.org/10.20783/DIAS.501. The corresponding author can also be contacted at vincent.humphrey@bluewin.ch. Source data are provided with this paper.

## Code availability

Code and documentation for CCSM4 is publicly available at https://www.cesm.ucar.edu/models/ccsm4.0/. Code and documentation for ECHAM6 (MPI-ESM) is available for scientific users at https://mpimet.mpg.de/en/science/modeling-with-icon/code-availability. Code and documentation for the GFDL model is publicly available at https://www.gfdl.noaa.gov/modeling-systems-group-public-releases/. Code and documentation for the IPSL model is publicly available at https://cmc.ipsl.fr/ipsl-climate-models/ipsl-cm5/. Model outputs were processed using the software Matlab 2019a.

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

**Acknowledgements** This research was funded by a Postdoc.Mobility fellowship of the Swiss National Science Foundation (P400P2_180784). C.F. acknowledges funding through NASA IDS grant 80NSSC17K0687. P.G. acknowledges funding from NASA 80NSSC18K0998 and European Research Council synergy grant USMILE ERC CU18-3746. P.C. acknowledges funding from the ANR CLAND convergence institute. S.I.S. acknowledges partial support from the European Union's Horizon 2020 Research and Innovation Program (grant agreement 821003 (4C)). We thank all modelling groups who participated in the GLACE-CMIP5 experiments and conducted the model runs, in particular F. Cheruy, S. Hagemann and D. Lawrence. We also thank G. Bonan, J. K. Green, M. Hirschi, D. Lawrence, D. Miralles, U. Weber and Y. Yin for comments on the analyses, data availability or the manuscript.

**Author contributions** V.H. designed and conducted the study. S.I.S. designed and coordinated the GLACE-CMIP5 climate model experiment. A.B., P.C., P.G., M.J., M.R., S.I.S. and C.F. provided feedback on the analyses, the figures and the manuscript.

**Funding** Open access funding provided by Max Planck Society.

**Competing interests** The authors declare no competing interests.

**Additional information**
**Correspondence and requests for materials** should be addressed to V.H.

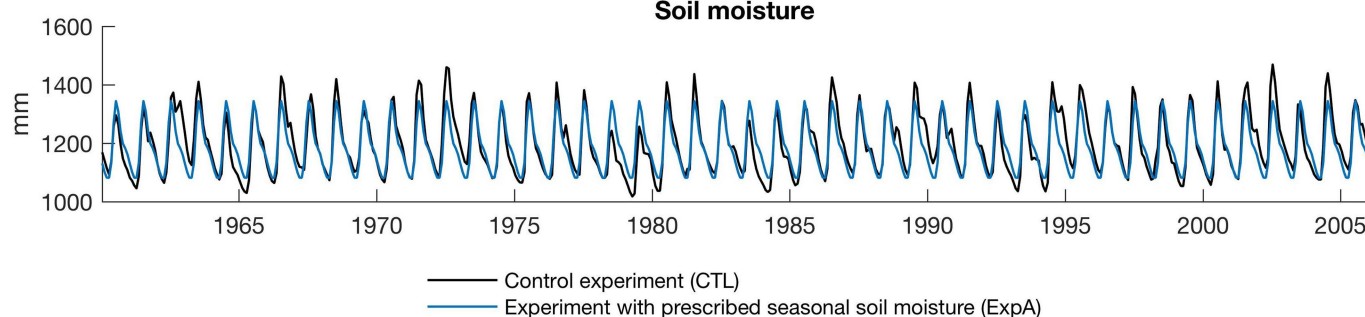

**Extended Data Fig. 1 | Soil moisture treatments in CTL and experiment A simulations.** At each grid point, the seasonal cycle calculated from the control experiment (CTL) is prescribed into the factorial experiment (ExpA). These example times series are taken from the CCSM4 model at 2° N and 58° W (northeast Amazon region).

Control simulation (CTL)

Experiment (ExpA)
- no inter-annual variability in soil moisture -
(only seasonal variability)

Atmosphere

Atmosphere

$\longrightarrow$ $T_{CTL}$ , $VPD_{CTL}$

$\longrightarrow$ $T_{ExpA}$ , $VPD_{ExpA}$

$SST_{CMIP5}$ $SM_{CTL}$

$SST_{CMIP5}$ $Seas(SM_{CTL})$

Ocean Land

Ocean Land

**Extended Data Fig. 2 | Concept of GLACE.** Setup of the control simulation (left) and the experiment with prescribed seasonal soil moisture (right). Sea ice, land use, and atmospheric $CO_2$ concentration also prescribed from CMIP5 in both experiments.

## $\Delta Q_{95}(T)$ when suppressing SM variability

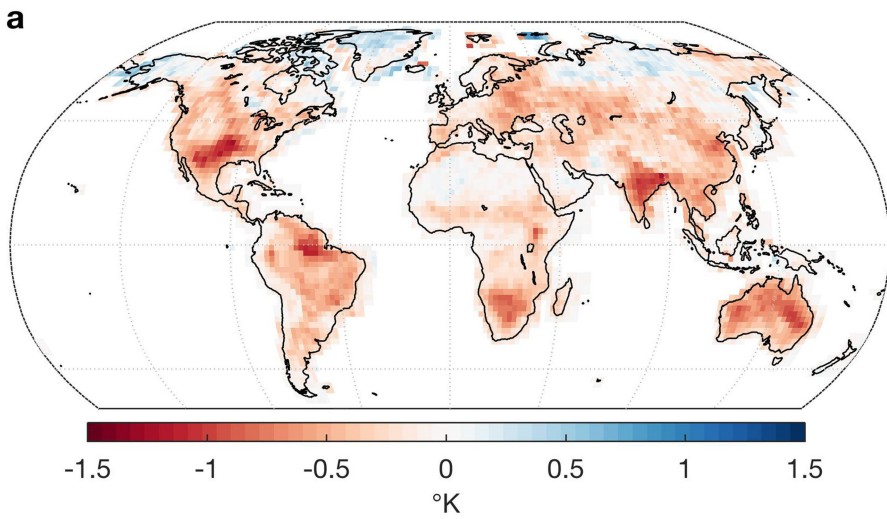

**a**

-1.5    -1    -0.5    0    0.5    1    1.5

°K

## $\Delta Q_{95}(VPD)$ when suppressing SM variability

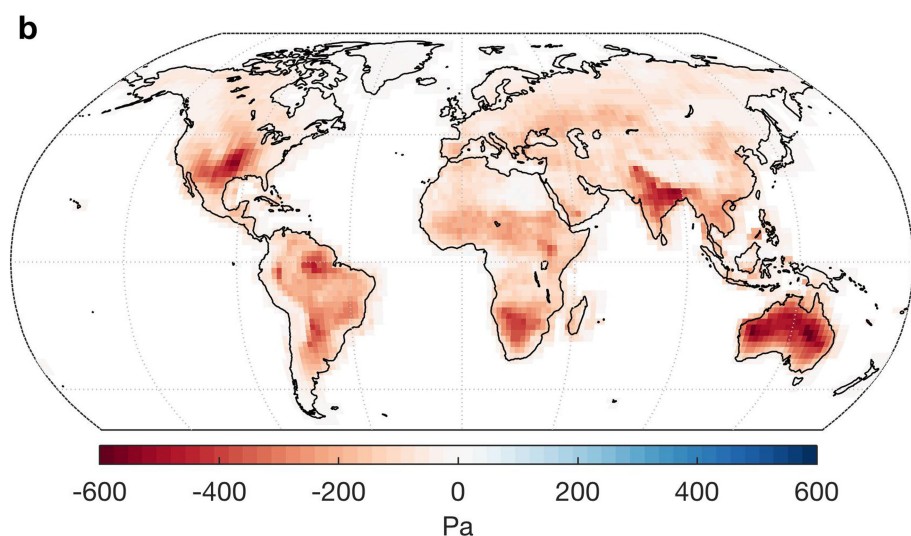

**b**

-600    -400    -200    0    200    400    600

Pa

**Extended Data Fig. 3 | Temperature and VPD extremes influenced by LAC.**
**a**, **b**, Change in the 95th percentile between the distributions of de-seasoned and de-trended temperature (**a**) and VPD (**b**) between CTL and experiment A

$(\Delta Q_{95} = Q_{95}^{ExpA} - Q_{95}^{CTL})$. The median $\Delta Q_{95}$ of all four models is reported. Suppressing non-seasonal soil moisture variability in experiment A reduces temperature and VPD extremes, demonstrating the role of LAC.

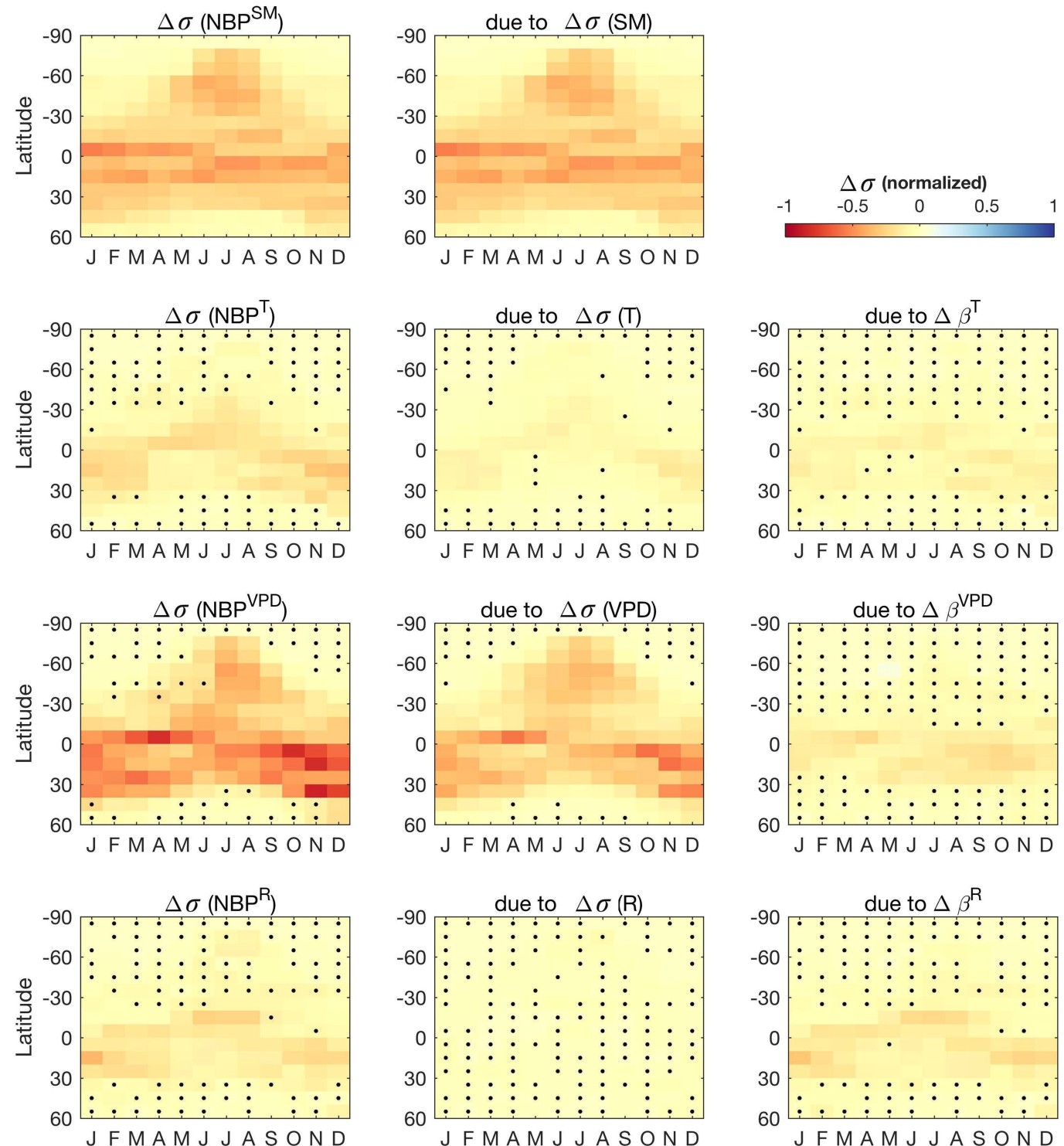

**Extended Data Fig. 4 | Change in annual NBP variability between CTL and experiment A.** Evaluation of the latitudinal change in NBP standard deviation ($\Delta\sigma$) between CTL and experiment A, decomposed by meteorological driver according to the sensitivity analysis. Negative values indicate a decrease of the NBP variability in experiment A compared to CTL. The middle and right columns indicate how much of this change is due to a change in the variance of the meteorological driver between experiment A and CTL, or due to a change in the sensitivity of NBP to that driver respectively (also see equation (1)). The results for each model are normalized by the model's NBP standard deviation (calculated across the entire space–time domain) and the median across models is depicted. Black dots indicate that at least one model disagrees on the sign of the change.

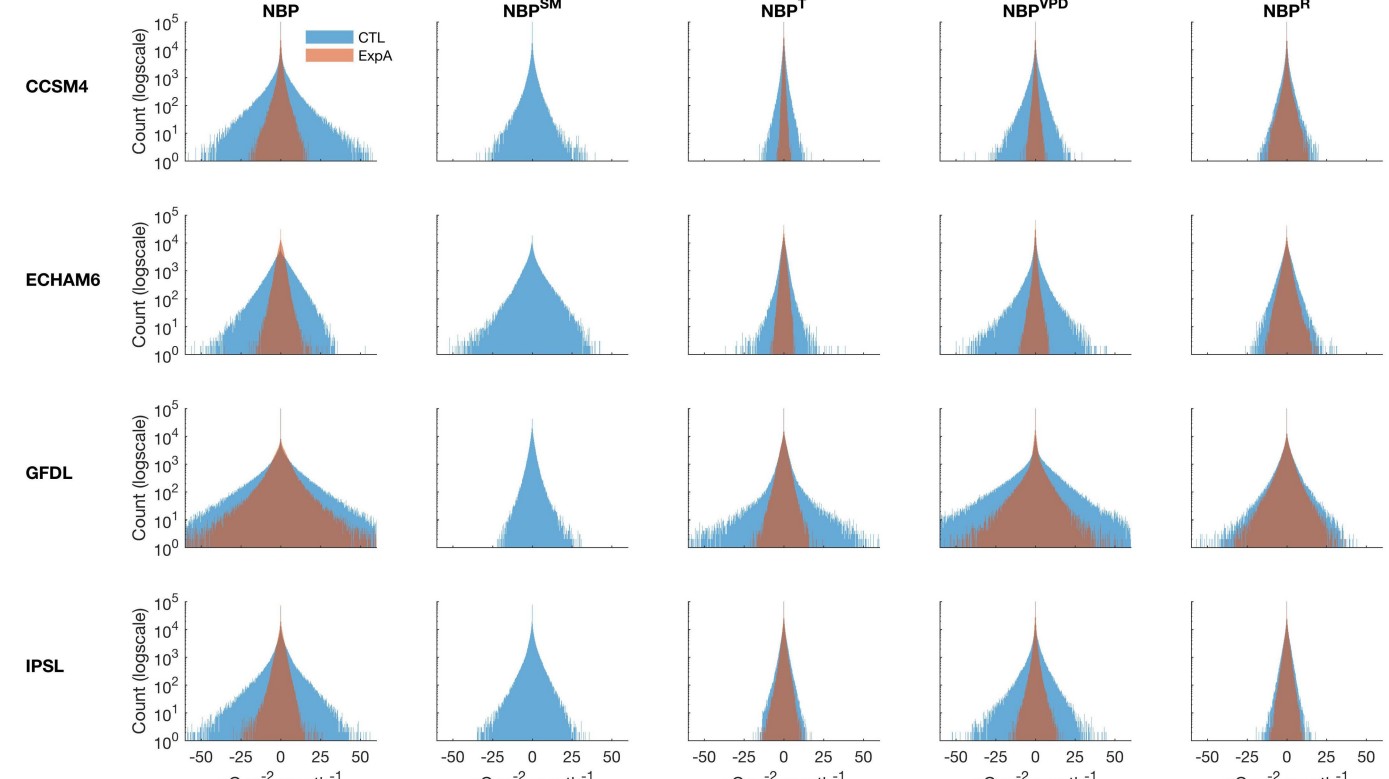

**Extended Data Fig. 5 | NBP anomalies in CTL and experiment A.**
Distributions (all grid points, at all time steps) of modelled NBP anomalies (left column) and their decomposition into meteorological drivers with the sensitivity analysis (other columns) for the control experiment (CTL) and the experiment with only seasonal soil moisture (ExpA). Rows correspond to each of the four climate models. Note the logarithmic scale on the vertical axis.

By construction, there are no soil moisture-driven NBP anomalies in experiment A in the second column (because seasonal soil moisture is prescribed in this experiment). The magnitude of the temperature-driven and VPD-driven NBP anomalies (third and fourth columns) is substantially reduced in experiment A (as a result of soil moisture–atmosphere feedback mechanisms).

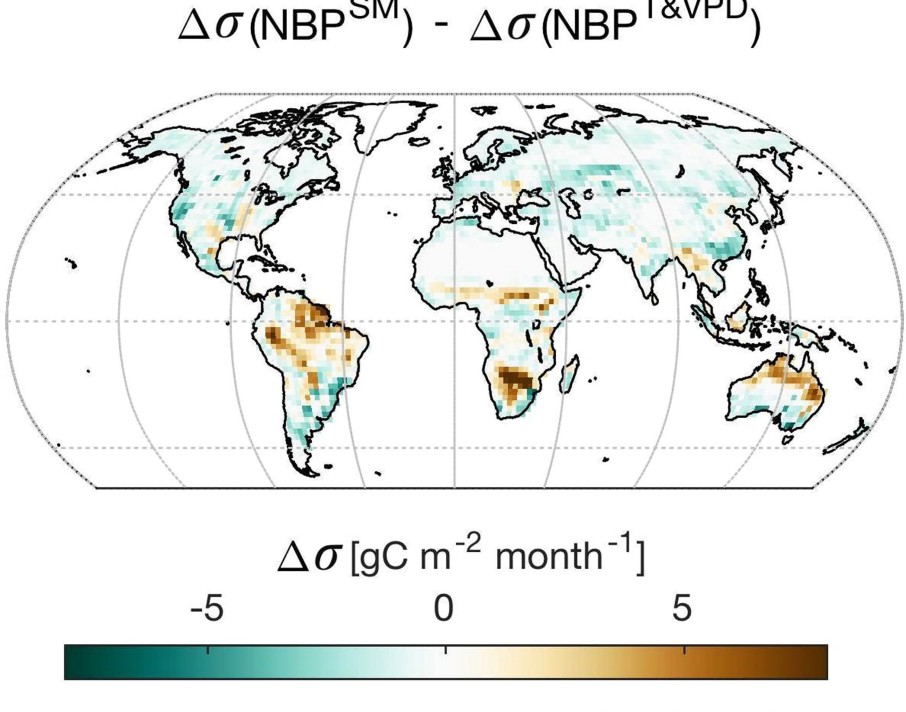

**Extended Data Fig. 6 | Comparison of direct versus indirect effects.** Difference between the magnitudes of direct effects (Fig. 2b) versus indirect (feedback) effects occurring through temperature and VPD (Fig. 2c).

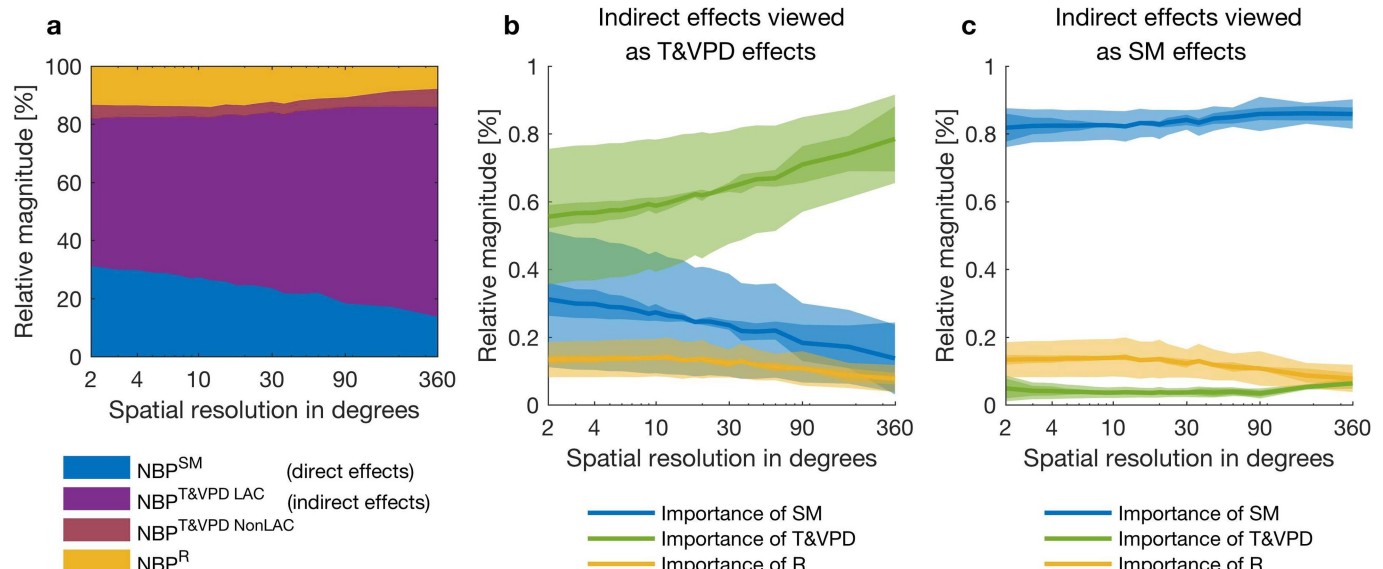

**Extended Data Fig. 7 | Opposing perspectives on drivers of NBP IAV reconciled by soil moisture–atmosphere feedback. a**, Relative magnitude of individual NBP components across spatial scales (same as Fig. 3b). **b**, **c**, The apparent relative importance of the meteorological drivers depends on how the indirect effects of soil moisture on temperature and VPD are viewed. Outer uncertainty bounds indicate the model spread (ensemble mean ±1$\sigma$), inner uncertainty bounds indicate the spread (±1$\sigma$) in the tendency (that is, the relative change from local to global scale; see Methods).

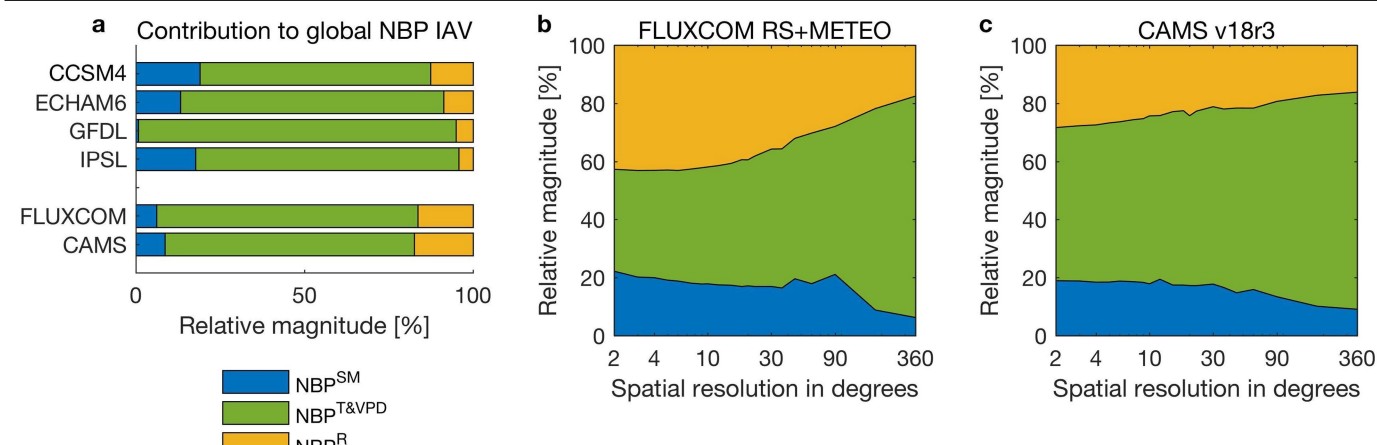

**Extended Data Fig. 8 | Sensitivity analysis compared to observational estimates. a**, Contribution of different meteorological drivers to global NBP IAV as estimated from the control simulations and from two independent observational products. Here, NBP$^{T\&VPD}$ is not separated into a LAC and non-LAC contribution as in Fig. 3b (because this cannot be done with the observational datasets). **b**, Same as Fig. 3b, but based on FLUXCOM. **c**, Same as Fig. 3b, but based on CAMS.

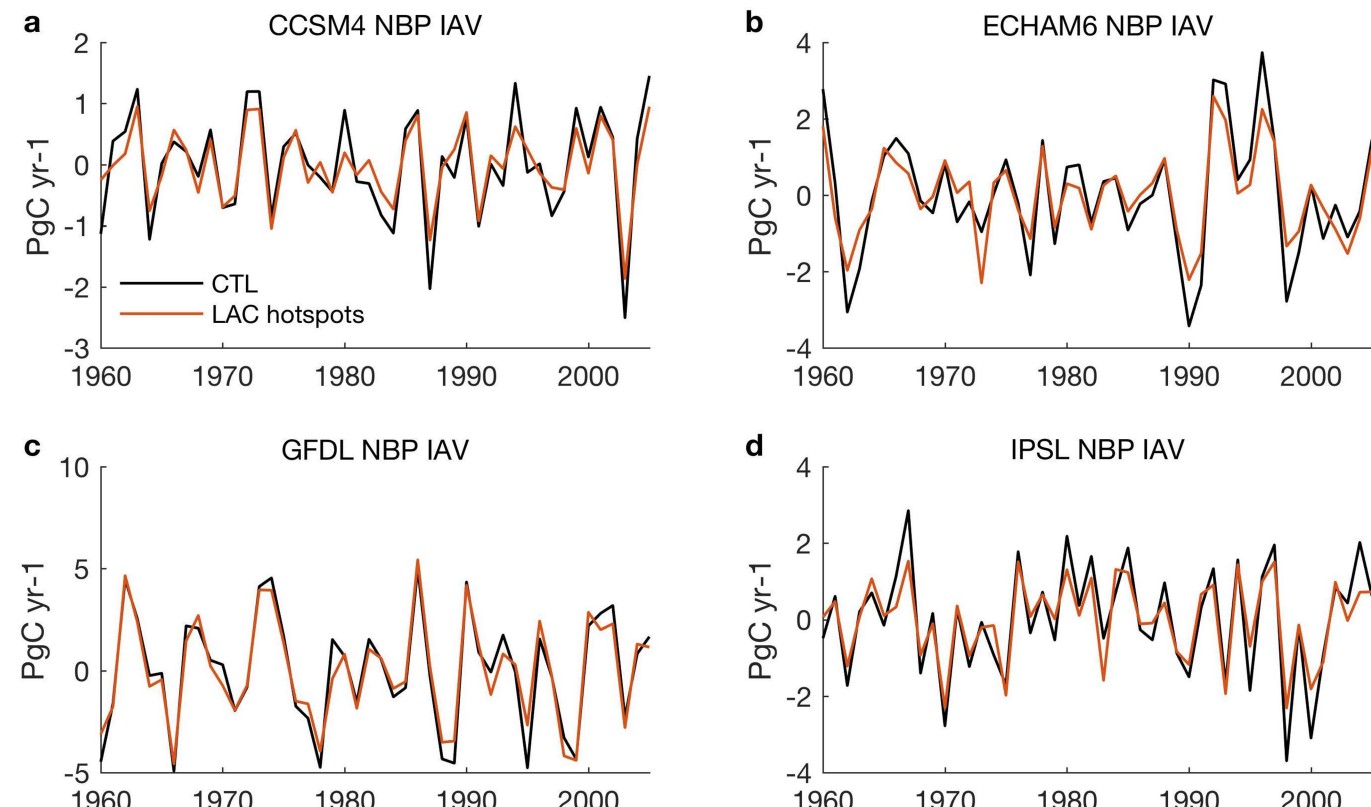

**Extended Data Fig. 9 | Contribution of LAC hotspots to global NBP IAV.** Global NBP IAV from the control experiment (CTL) calculated over all land grid cells versus only over the LAC hotspots identified in Fig. 4.

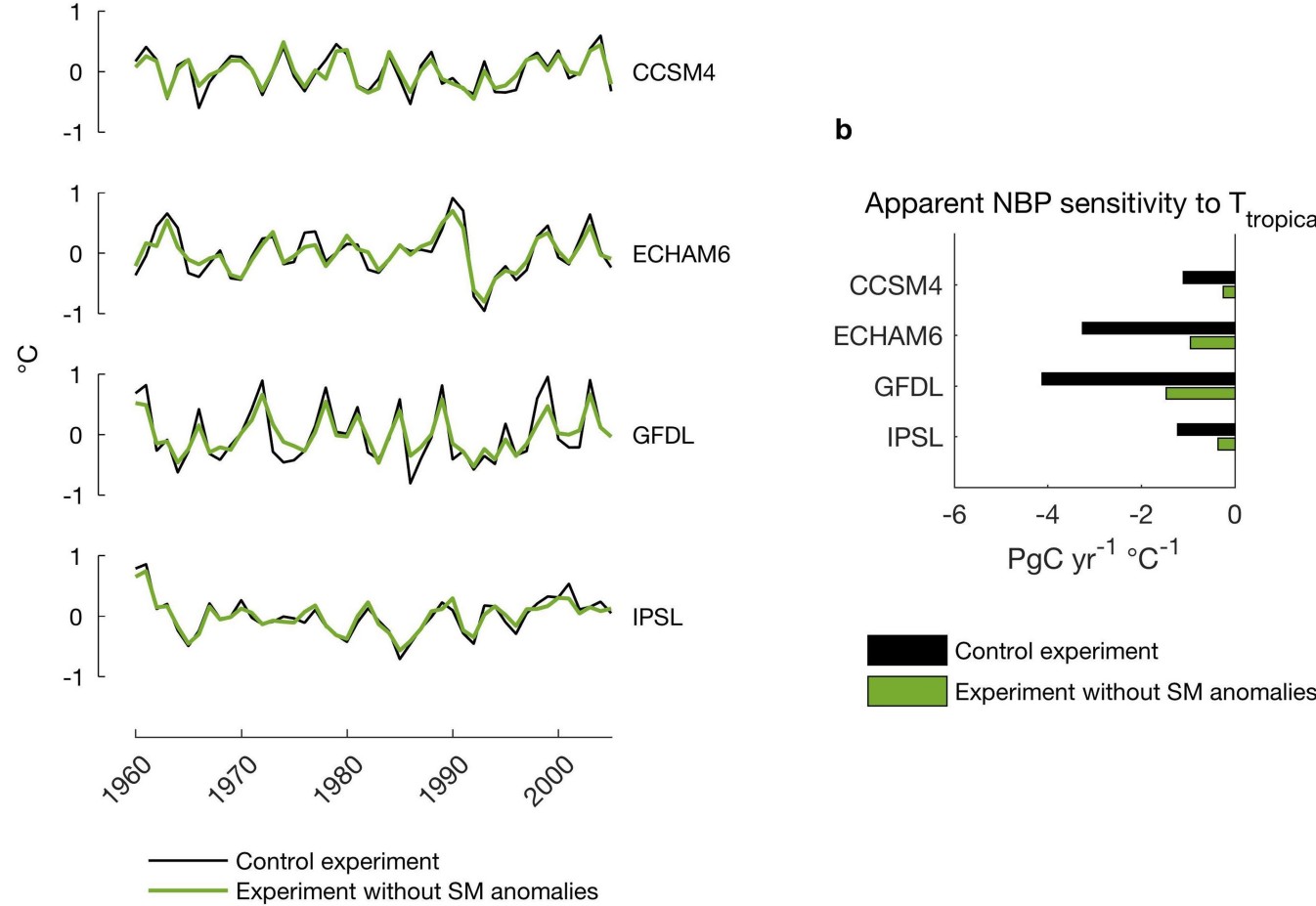

**a**, Inter-annual variability in tropical mean temperature

**b**, Apparent NBP sensitivity to T$_{tropical}$

**Extended Data Fig. 10 | Tropical temperature in CTL versus experiment A.**
**a**, IAV in tropical (24° N to 24° S) mean land temperature in model experiments with and without variability in soil moisture (similar to Fig. 1a for NBP). **b**, Apparent sensitivity of global mean NBP to tropical mean temperature in CTL and experiment A.