## [Peer Review File · Nature]

Manuscript Title: Soil moisture–atmosphere feedbacks dominate land carbon uptake variability

Reviewer Comments & Author Rebuttals**Reviewer Reports on the Initial Version:****Ref #1**

This is an elegant model analysis that nicely clarifies the interactions among soil moisture, temperature and VPD in determining interannual variability in the land C uptake. I found the work to be insightful, but not completely convincing, or completely clear. Some parts could be re-written for additional clarity. I would also suggest a few extra plots would help to really nail down the story.

The methods description of the model simulations was a bit too brief for my liking. In particular, I would have liked to understand what forcings are used to drive the model (so that I could gain a better understanding of what drives the IAV in the models). Is it the case that the control simulations are driven only by CO₂, SST, sea ice and land use? Such that most of the IAV is driven by SST? To what extent are there inherent dynamics in the models that lead to IAV? Some brief background in the introduction about the source of IAV in the models would be helpful to interpret the results.

I assume that when ExpA is run, there is no longer mass balance of water. It would be useful to understand what the implications are for precipitation and cloud cover. (NB line 282 – perhaps the only difference in the forcings – not the only difference between the simulations!)

It would be particularly helpful to see the interannual variability in temperature & VPD occurring in the two sets of model runs. The argument is made that much of the IAV in NBP is driven by soil moisture impacts on T & VPD – but we don't actually get to see how large that effect is. Plots of IAV in T, VPD and radiation similar to Fig 1 would be great to help convince that this is the right interpretation of what is happening.

Figure 4 does show IAV in tropical mean T, but it seems that the soil moisture does not actually lead to large changes in tropical mean T – which tends to undermine the conclusion that it is changes in T and VPD due to differences in soil moisture that cause most of the IAV. The explanation given is that it is the extremes of temperature, not the mean temperature, that are important. This is described in the para starting line 188. Unfortunately, this paragraph is quite hard to follow, and doesn't present all the evidence or information that it needs to. For example, this line: "This is because suppressing soil moisture anomalies has little impact on the temperature mean, even though it strongly reduces the extremes in temperature (and VPD) that cause most of the NBP variability." How do we know that the extremes have been reduced? How can we be sure that it is the extremes in T and VPD that cause the variability?

The analysis that shows the sensitivity of IAV to anomalies needs to explain what is meant by "an" anomaly – is it on a grid point basis, monthly basis?? Some mention in the methods would be appropriate. Is it a problem for the analysis that modelled NBP is so sensitive to extremes? Can the linear regression approach be justified if there is high sensitivity to extremes?

I also struggled to follow what was being shown in Figure 3 (esp c-d). What is meant by the "uncoupled perspective" in Fig 3c? I spent a bit of time wondering if uncoupled models had been run for this. Some simpler description of what is shown would be warranted.

Finally, I would have liked to see some text discussing the fact that the work is entirely based on models – we need to be a bit cautious about over-interpreting the models. To what extent do the CTRL simulations actually capture real IAV, for example? Is the modelled IAV in temperature, VPD, soil moisture and NBP realistic?

Ref #2

Dear authors

With pleasure I've read this model study. The paper is clearly written and with the SI I can understand and follow the model performance and the model experiments. Although the paper gives a clear conclusion, clearly tested with the model experiments, I'm not convinced that this work is ground breaking. In a way it is understanding why the models behave differently in coupled and uncoupled conditions. It doesn't tell anything about the real mechanisms and the consequences of the carbon cycle.

1. In the summary it is unclear what exactly the role is of different vegetation and the length of the drought. It is not new that soil moisture has a large impact on photosynthesis and that there is a strong feedback to the atmosphere.

2. L44: observations from networks and satellites: Indeed this is very important information and there is a quite a bit of work currently done on how vegetation responds. Hot debates are the greening and browning and the effect on the carbon cycle. These greenings can be a result of anthropogenic forcings (revegetation) and as a result of cascading effects due to atmospheric moisture transport. However, I don't see any comparison between observations and the current models, so we don't know if the change in variability is large compared to the mismatches between observations and models. The paper shows only comparisons between model experiments. For me it hereby misses the significance of the work in terms of mechanisms and observed trends in current vegetation.

3. L69: the use of stress functions in these kind of models is debated. Alternative is via limitations (optimality). Largest differences between those approaches will be found during extremes (droughts). Will it be that the consequence of using limitation functions will have a higher effect on the variability? This isn't an easy debate. It is the question if the current land surface models can be used to ask these fundamental mechanistic questions.

4. L102: 1). "In other words, without SM variability, the interannual variability (IAV) of land carbon uptake is almost eliminated". This isn't new or surprising. Drought plays a crucial role in photosynthesis. Further, is it that soil moisture anomalies will have highest effect on inter-annual variability. Higher than temperature for instance?

5. Figure 1: although reduction in standard deviation, there is still a large difference between the models. Is it so that the variability between the models was and is still bigger than the variability within the model? And if so, what should we conclude from it? And second, I also would like to know what happens with the average fluxes, not only the variability. If the average fluxes of carbon gain and losses have large deviations between the models and between the observations, is this then really the big issue to discuss?

6. L229: I think so this is a crucial part in the discussion. Indeed all models are parameterized and different parameterizations could might work better in coupled and uncoupled ways. But focus on the best estimate doesn't necessary help us in understanding the mechanisms. Besides the earlier mentioned debate on limitation versus stress function of course also the biome approach compared to the trade approach.

7. Figure 2: Why is it that the Congo basin is reacting differently compared to the other tropical forests? (Amazon, Asia)

Ref #3

The manuscript describes a modelling effort to ascertain how much variation in net carbon dioxide uptake by the terrestrial biosphere is associated with variation in soil moisture content directly, and how much is due to indirect effects via the associated reduction in evaporation, causing a drying and heating of the air passing across the vegetation. It is a useful contribution as unfortunately some scientists have ascribed effects in off-line models to, for example, an increase in warming, without recognizing that in the runs examined the warming was actually a response to reduced rainfall. This idea was well expressed earlier, and deserves quotation here, by Yin, D., Roderick, M.L., Leech, G., Sun, F. and Huang, Y., 2014. The contribution of reduction in evaporative cooling to higher surface air temperatures during drought. *Geophysical Research Letters*, 41(22): 2014GL062039.

The present manuscript is careful to acknowledge that rainfall itself will be affected by changes in air flow patterns.

It is a little odd that soil moisture is separated from VPD and temperature, but the latter are not separated. Why not? Giving the responses to temperature, separated from VPD and soil moisture, would expose the modelled effect of temperature. This would be helpful, as it has tended to be lost in the complex of equations necessary to describe biosphere functioning.

I realise that it is not my job to comment on the English, but there are a couple of places in the introduction where it is a little obscure.

Line 22 "changes in carbon uptake by terrestrial ecosystems play an essential role for atmospheric CO₂ concentrations." I suggest: "changes in carbon uptake by terrestrial ecosystems play an essential role in determining atmospheric CO₂ concentrations".

Line 31 explain "coupled" when it first appears.

Line 35 "is often studied by intervening solely on soil moisture content". How about "is often studied by intervening solely IN soil moisture content"

A more important point is the final conclusion at lines 37 and 38, which should read: "Our results suggest that most of the variability in MODELLED global land carbon uptake is driven by temperature and vapour pressure deficit effects which are controlled by soil moisture." The 'modelled' is important as the observation that 60% of the effects come indirectly is totally sensitive to the modelling of vegetation responses. The experimental results of Wong et al. (Nature) had a big impact via the Ball-Berry equation linking stomatal conductance with relative humidity. The equations fitted well generally and together with slight modification by Leuning became very popular. There was nothing quantitatively equivalent in terms of the effects of soil moisture content. So the balance of impact has tended to be described via the easily measurable responses of gas exchange to VPD. My point is that the 60% value is a very soft, modelled one.

Line 43 Reference 4 is not really a theoretical advance. It is a convenient fix which assumes that leaves are always RuBP regeneration limited (when in fact in sunlight they are not), introduced to give the correct sense of response of stomatal conductance to changes in [CO₂].

Line 65. Susceptible to having

I congratulate the authors.

Ref #4

This paper uses results from coupled land-atmosphere simulations with a GCM ensemble to diagnose direct and indirect contributions of soil moisture to NBP variability of land ecosystems globally and regionally. By comparing a fully-coupled control simulation with a simulation in which an annual cycle of soil moisture is prescribed from climatology in each grid cell, the study infers that interannual soil moisture variability is the dominant control on variation in NBP. Indirect effects mediated by effects of surface evaporation on near-surface temperature and humidity are found to be stronger than the direct effects of soil water availability on photosynthesis. Tropical

and semi-arid biomes are identified as 'hotspots' for this driving pattern.

This study is a contribution to a running debate on the climatic controls of NBP variability in low latitude ecosystems, which in turn control interannual variations (IAV) in atmospheric anomalies of CO₂. A number of high-profile papers have been published, some asserting soil moisture control and others temperature control, the latter purportedly mediated by effects on respiration. The cited paper by Shilong Piao and coauthors (2019) reviewed most of the key papers and employed structural equation modelling to conclude that interactions between temperature and moisture availability, that is, the dependence of carbon cycle sensitivity to temperature on moisture conditions, was the dominant control of IAV. I think this is a realistic framing, which goes some way towards reconciling apparently conflicting findings of different studies as to the controlling drivers. In general studies that identify temperature as the key driver come to this conclusion based on a simple correlation of NBP or a proxy such as RLS, IAV etc on large-scale climate (Anderegg 2015, Cox 2013). Bottom-up studies based on local flux observations or process modelling (Jung 2017, Ahlström 2015) tend to unearth more subtle causal relationships in which hydrological variations and large-scale circulation phenomena such as ENSO play an underpinning role.

The present paper adds to this existing tapestry of studies. Results are revealing but not unexpected given the known behavior of the models. The paper is the latest in a series based on the GLACE-CMIP5 ensemble, here comprising a subset of four CMIP5-generation GCMs. It should be noted that these models are now being superseded by a new generation being deployed for CMIP6, in many cases adopting more sophisticated schemes for land surface dynamics and biogeochemistry. That said, I believe the general behavior and driver sensitivity of the models will not have changed so much that the conclusions of this paper would change, were the analysis to be repeated using the latest ESM versions of the same models.

The paper has some similarity with Green et al (2019) also in Nature which used the GLACE-CMIP5 ensemble to highlight the role of soil moisture as a driver of NBP evolution in the context of an RCP8.5 21st century climate-emissions scenario. The present paper demonstrates high NBP sensitivity to soil moisture is upheld from multi-annual to decadal-centennial time scales. This will not necessarily be true of the real Earth system. Disregarding feedbacks to climate through the carbon cycle (not accounted for in either paper) there are still several reasons why we might expect shifts in the sensitivity to environmental drivers as we move up to longer time scales. A shift in NBP will affect the amount of respiring live biomass and dead organic matter over time. Shifts in vegetation demography and PFT distributions might affect phenology, hydraulics and other vegetation properties, in turn impacting productivity directly and via evapotranspiration feedback. The GLACE ensemble only incorporates the first item from this list (changing substrate for respiration), and it turns out this is not an important aspect at multi-annual scale, with the response of GPP dominating the response of NBP (in line with e.g. the Piao et al analysis). While this paper and Green et al. form a useful companion set, conclusions are largely informed by the same environmental controls and biophysical feedback mechanisms playing out in the models. In my opinion, this is an issue for rating the novelty of the present paper.

I believe the inferential approach of the study, comparing a coupled and uncoupled model experiment (wrt soil moisture) is sound. However, I think the narrative claiming that the coupled land-atmosphere modelling reveals something new, not captured by other studies in this space, is overstated. The feedbacks that are explicit in a coupled model are implicit in the observations entering analyses such as Jung 2017, using upscaled flux tower data. In the case of offline modelling, the forcing data on temperature, rainfall etc incorporate real-world dependencies on soil moisture, evapotranspiration and energy balance. So long as the offline model has some skill in reproducing vegetation properties that control these quantities, and the quantities themselves, there is no reason to expect the influence of this coupling on the relationship between soil moisture and NBP would be missed. The main advantage of a coupled over an uncoupled approach is the ability to partition the overall relationship between a direct and coupled effect. I do not

agree with the assertion on L170-173 that this quality of offline models to capture the full response will somehow break down when forced by GCM output fields in a future climate projection. The statement suggesting this refers to Figure 3c as supporting evidence, but here the coupling between soil moisture and air temperature and VPD was deliberately broken when the models were forced by a climatology of soil moisture from the control run. This will not be the case when output from a straight GCM projection (i.e. with physically consistent links between different output fields) is used as input to an offline model simulation.

The paper does make some good points about limitations of current land surface models that can be traced to their history (L230-237). Both coupled and offline models incorporate stomatal conductance parameterisations that capture empirical dependencies on temperature and VPD/humidity but there is no established standard for capturing sensitivity to soil moisture, with a range of approaches, ranging from simple scalars to complex, e.g. optimisation-based, approaches being deployed. A number of models substitute a direct dependence of rubisco capacity for which evidence is sparse, except under extreme conditions. However these criticisms apply equally to the vegetation schemes incorporated in coupled ESMs, which are usually a rewired version of an offline LSM or DGVM.

In summary, I find this a sound and well-presented paper that makes a useful contribution to an already fairly substantial body of global-scale studies seeking to identify the causal mechanisms behind biospheric control of NBP/IAV. Results appear to be consistent with Piao et al (2019) who arrived at the same conclusions using a combination of approaches, and also relates to the Green et al. (2019) study using the same model ensemble to address the longer-term sensitivity of NBP to its candidate drivers. The cited references make appropriate credit to previous work.

Author Rebuttals to Initial Comments:

This document contains:

1. Overview of major changes
2. Detailed response to referees' comments and suggestions

1. Overview of major changes

1. **Evaluation of modelled carbon fluxes.** We have conducted a comprehensive evaluation of the ability of the models to adequately represent real-world variability in land carbon uptake. We show that the simulated spatial patterns of NBP variability are in reasonable agreement with observational estimates from two independent sources (Supplementary Figure S13, L149-150, L277-292). Even though there are some differences in spatial patterns and amplitudes, models agree with the observational estimates to the same extent that these (also uncertain) observations agree with each other. We also show that the model-based attribution of NBP variability to the different meteorological drivers is consistent with what can be inferred from observational datasets alone (Supplementary Figure S12, L146-148, L379-394). Finally, we show that the simulated spatial patterns and global amplitudes of GPP variability agree well with

machine-learning-based and satellite-based estimates (Supplementary Figure S21, L299-305).

- 2. Evaluation of the main climatic drivers.** In addition to evaluating the models' representation of land carbon fluxes, we also verify that the models are able to reasonably reproduce real-world variability in soil moisture, temperature and vapour pressure deficit (Supplementary Figures S18-20, L294-299).
- 3. Spatial consistency between land-atmosphere coupling hotspots and NBP variability.** Figure 4 of the main text was moved to Supplementary Figure S15. The new Figure 4 shows that regions of strong land-atmosphere coupling – where temperature and VPD variability is most affected by soil moisture feedbacks – are spatially consistent with regions of large NBP variability. This provides an additional perspective to our main finding that indirect temperature and vapour pressure deficit effects controlled by soil moisture drive most of the inter-annual variability in land carbon uptake.

2. Detailed response to referees' comments and suggestions

Referee #1 (Remarks to the Author):

This is an elegant model analysis that nicely clarifies the interactions among soil moisture, temperature and VPD in determining interannual variability in the land C uptake. I found the work to be insightful, but not completely convincing, or completely clear. Some parts could be re-written for additional clarity. I would also suggest a few extra plots would help to really nail down the story.

The methods description of the model simulations was a bit too brief for my liking. In particular, I would have liked to understand what forcings are used to drive the model (so that I could gain a better understanding of what drives the IAV in the models). Is it the case that the control simulations are driven only by CO₂, SST, sea ice and land use? Such that most of the IAV is driven by SST? To what extent are there inherent dynamics in the models that lead to IAV? Some brief background in the introduction about the source of IAV in the models would be helpful to interpret the results. I assume that when ExpA is run, there is no longer mass balance of water. It would be useful to understand what the implications are for precipitation and cloud cover. (NB line 282 – perhaps the only difference in the forcings – not the only difference between the simulations!)

Thank you very much for the positive review. This is a good point. We have expanded the description of the model experiment in the Methods section to answer these questions more thoroughly. We now discuss other internal sources of variability in the two experiments and the impact on the water balance at lines L245-261.

It would be particularly helpful to see the interannual variability in temperature & VPD occurring in the two sets of model runs. The argument is made that much of the IAV in NBP is driven by soil moisture impacts on T & VPD – but we don't actually get to see how large that effect is. Plots of IAV in T, VPD and radiation similar to Fig 1 would be great to help convince that this is the right interpretation of what is happening.

Thank you for this comment. We realize that a depiction of SM impacts on T&VPD itself was lacking. Figure 4 has been modified to better convey this point and now shows the effect of prescribing SM on T & VPD IAV. We also show the effects of prescribing SM on T & VPD extremes specifically in Supplementary Figure S3. As requested, we also provide plots of the global mean IAV of T, VPD and radiation similar to Fig.1 in Supplementary Figure S17.

Figure 4 does show IAV in tropical mean T, but it seems that the soil moisture does not actually lead to large changes in tropical mean T – which tends to undermine the conclusion that it is changes in T and VPD due to differences in soil moisture that cause most of the IAV. The explanation given is that it is the extremes of temperature, not the mean temperature, that are important. This is described in the para starting line 188. Unfortunately, this paragraph is quite hard to follow, and doesn't present all the evidence or information that it needs to. For example, this line: “This is because suppressing soil moisture anomalies has little impact on the temperature mean, even though it strongly reduces the extremes in temperature (and VPD) that cause most of the NBP variability.” How do we know that the extremes have been reduced? How can we be sure that it is the extremes in T and VPD that cause the variability?

Thank you for this very useful comment (note that Figure 4 of the original submission was moved to Supplementary Figure S15 and replaced by a new figure that better explains this point). We realize that we did not clearly explain that the reduction of T and VPD variability due to prescribing soil moisture is not occurring everywhere but only over land-atmosphere coupling hotspots (this is now shown clearly in Figure 4b-c, Supplementary Figure S3, and explained at L169-172). Because these reductions have a limited spatial extent, the overall mean of tropical land temperature (now in Supplementary Figure S15) is not greatly affected.

Because regions of strong land-atmosphere coupling (where the variability in T and VPD is most reduced when soil moisture is prescribed) also correspond to regions where NBP variability is the largest (Figure 4d, Supplementary Figure S14), global mean NBP is greatly affected, even though tropical mean T does not change much.

One important take-away from this finding is that the sensitivity of global NBP IAV to tropical mean temperature (Supplementary Figure S15b), which is reported and used in numerous studies, might not accurately reflect the mechanism at play. The experiment shows that most of the NBP variability vanishes when prescribing SM, even though tropical mean T is left almost unchanged. If NBP IAV was actually sensitive to tropical mean temperature, this could never happen. This is what we

wanted to illustrate in Figure 4 of the original submission, now in Supplementary Figure S15 and explained at lines L172-176.

The analysis that shows the sensitivity of IAV to anomalies needs to explain what is meant by “an” anomaly – is it on a grid point basis, monthly basis?? Some mention in the methods would be appropriate. Is it a problem for the analysis that modelled NBP is so sensitive to extremes? Can the linear regression approach be justified if there is high sensitivity to extremes?

Thank you very much for this comment. In order to focus on IAV, we calculate anomalies by subtracting the seasonal cycle and the long-term trend. This is now clearly stated in the main text at L85-86.

The regression analysis is indeed done on a grid-point, monthly basis. Because the regressions are done month-by-month, the overall approach is actually able to accommodate a fair amount of non-linearity (as the sensitivities are local and allowed to vary for each month). That said, we fully recognize that some amount of variability won't be reproduced with this approach, partly because of potentially non-linear responses within a given month (and also, memory effects). This is discussed at L373-378. The regression is evaluated with Supplementary Figures S4-S6. As stated in the Methods, the quality of the regression varies depending on the model, but works well overall. In particular, the impact of prescribing soil moisture is well captured by that regression (Supplementary Figure S7, L396-403). We believe that the current method is a suitable compromise between allowing some degree of non-linearity while still using a reasonable number of regression parameters.

I also struggled to follow what was being shown in Figure 3 (esp c-d). What is meant by the “uncoupled perspective” in Fig 3c? I spent a bit of time wondering if uncoupled models had been run for this. Some simpler description of what is shown would be warranted.

First, we note that subfigures 3c-d appear now in Supplementary Figure S10. We agree that this section was confusing and have improved the manuscript. The reviewer is correct that there are no uncoupled runs in this paper.

What we want to show is that our results reconcile opposing perspectives on the roles of temperature versus water availability. The apparent importance of either driver depends on whether the indirect effects are viewed as controlled by temperature or soil moisture (see Supplementary Figures S10-S11). This is now better explained at lines L143-146 and we have updated the figure labels to convey this more clearly.

Finally, I would have liked to see some text discussing the fact that the work is entirely based on models – we need to be a bit cautious about over-interpreting the models. To what extent do the CTRL simulations actually capture real IAV, for example? Is the modelled IAV in temperature, VPD, soil moisture and NBP realistic?

Thank you for this comment. We agree that a comparison with observed IAV was necessary. As suggested, we evaluate simulated IAV of NBP, soil moisture, temperature, and VPD against available observations in Supplementary Figures S13 and S18-20. For NBP in particular, we note that while observational estimates of NBP variability exist, they are also fairly uncertain and do not agree perfectly with each other, reflecting our limited knowledge of carbon fluxes globally. Still, we show that although the models exhibit some differences, they agree with the observational estimates to the same extent that the observational estimates themselves agree with each other (Supplementary Figure S13g). This evaluation is discussed in the main text at lines L149-150 and an in-depth discussion is provided in the Methods section at L277-292. In the new Supplementary Figure S12, we also show that the models attribute about the same amount of variability to the different meteorological drivers as what is suggested by the observational estimates (L146-148 in the main text, L379-394 in the methods).

Referee #2 (Remarks to the Author):

Dear authors

With pleasure I've read this model study. The paper is clearly written and with the SI I can understand and follow the model performance and the model experiments. Although the paper gives a clear conclusion, clearly tested with the model experiments, I'm not convinced that this work is ground breaking. In a way it is understanding why the models behave differently in coupled and uncoupled conditions. It doesn't tell anything about the real mechanisms and the consequences of the carbon cycle.

1. In the summary it is unclear what exactly the role is of different vegetation and the length of the drought. It is not new that soil moisture has a large impact on photosynthesis and that there is a strong feedback to the atmosphere.

Thank you very much for your positive feedback. We agree that soil moisture is known to impact GPP and that the soil moisture feedback to the atmosphere can be important. However, we believe that this analysis is the first one (to our knowledge) to demonstrate the global relevance of soil moisture and its feedback on the atmosphere for global NBP inter-annual variability. Our contribution is also the first one (to our knowledge) to disentangle the relative global contributions of direct soil moisture effects versus the indirect response of the carbon fluxes to the atmospheric feedback. It is not trivial that the response to the atmospheric feedback alone could be larger than the response to soil moisture itself (Figure 2d) in several locations of the world which are notoriously under-observed by current in-situ networks. Because soil and atmospheric dryness do not equally respond to climate change, disentangling these direct and indirect effects seems a useful contribution and improves our overall understanding of what mechanisms drive the year-to-year variability of land carbon uptake. Finally, our analysis shows that changes in the sensitivity of yearly carbon uptake to mean tropical temperature might not be a reliable metric, precisely because it does not reflect the actual mechanisms revealed by the study. Because this sensitivity is routinely used to constrain projections of future carbon fluxes and

evaluate carbon cycle models, we believe our findings bring novel and impactful information.

2. L44: observations from networks and satellites: Indeed this is very important information and there is a quite a bit of work currently done on how vegetation responds. Hot debates are the greening and browning and the effect on the carbon cycle. These greenings can be a result of anthropogenic forcings (revegetation) and as a result of cascading effects due to atmospheric moisture transport. However, I don't see any comparison between observations and the current models, so we don't know if the change in variability is large compared to the mismatches between observations and models. The paper shows only comparisons between model experiments. For me it hereby misses the significance of the work in terms of mechanisms and observed trends in current vegetation.

We fully agree that an observational perspective was missing in the paper and have added several figures comparing the simulated inter-annual variability against the observed one (Supplementary Figures S13 and S18-20, now discussed at lines L149-150, L277-299). In short, while some models agree better with observational estimates (themselves also uncertain) than others, all models are unanimous in their evaluation of soil moisture impacts on NBP variability (that is, they all show a 90% reduction in NBP variability when direct and indirect soil moisture effects are suppressed, as shown in Figure 1 and Supplementary Table 2). In addition, we find that models and observations agree reasonably well on their attribution of global NBP variability to the different meteorological drivers (Supplementary Figure S12, L379-394).

We also agree with the reviewer that long-term trends in greenness and their effects on the carbon cycle are essential and current topics. However, we note that this study focuses on the inter-annual variability of NBP (not on the long-term trends). In fact, long-term trends are removed when calculating the anomalies.

3. L69: the use of stress functions in these kind of models is debated. Alternative is via limitations (optimality). Largest differences between those approaches will be found during extremes (droughts). Will it be that the consequence of using limitation functions will have a higher effect on the variability? This isn't an easy debate. It is the question if the current land surface models can be used to ask these fundamental mechanistic questions.

Yes, this is a very interesting debate, optimality approaches might potentially yield different results, especially with respect to long-term trends. Stress functions are parameterized to reflect the response of a given type of vegetation to an average climate. But if climate changes so will the vegetation, and as a result it might evolve to become more or less sensitive to certain types of environmental stressors, making it difficult to extrapolate their impact in the long run.

Here, as we focus on year-to-year variability over the historical period, this is slightly less of an issue, but still important because the stress functions determine which environmental factor is causing the variability. As mentioned above, our comparisons with observations (added in the revisions), are in broad agreement with what models

indicate as the dominant drivers of NBP variability (Supplementary Figure S12). This is from two completely independent observational sources (the FLUXCOM statistical upscaling of flux tower data and the CAMS atmospheric CO₂ inversion).

Finally, we note that the mechanism discussed in our study is also relevant for optimality-based approaches. At the ecosystem-scale, an optimum would likely not only depend on the soil moisture availability, but also on the feedback to the atmosphere. Our results suggest that because land-atmosphere coupling is occurring, finding an optimum response with respect to soil moisture availability also requires taking into account the resulting feedback on temperature and VPD.

4. L102: 1). “In other words, without SM variability, the interannual variability (IAV) of land carbon uptake is almost eliminated”. This isn’t new or surprising. Drought plays a crucial role in photosynthesis. Further, is it that soil moisture anomalies will have highest effect on inter-annual variability. Higher than temperature for instance?

We believe this is in fact a novel finding. In particular, the result that direct soil moisture impacts are minor relative to the impacts of atmospheric feedbacks to temperature and VPD is new. The experience shows that if ecosystems always had access to climatologically normal amounts of water (i.e. no droughts, nor excessive soil moisture), the global NBP variability would become 90% lower, but mainly because the atmospheric conditions would have less extremes, and not so much because plants would avoid soil water stress (according to the models). We believe this is an important finding, and also one that cannot be easily made from in-situ experiments since it involves atmospheric feedbacks that unfold at a quite large spatial scale. Also, we note that the study does not focus on SM impacts on photosynthesis (GPP) but on the net carbon balance (NBP).

5. Figure 1: although reduction in standard deviation, there is still a large difference between the models. Is it so that the variability between the models was and is still bigger than the variability within the model? And if so, what should we conclude from it? And second, I also would like to know what happens with the average fluxes, not only the variability. If the average fluxes of carbon gain and losses have large deviations between the models and between the observations, is this then really the big issue to discuss?

Yes, we agree that the models in Figure 1b disagree with each other in terms of their overall NBP IAV amplitude. We find that the standard deviation of global mean NBP from the different models ranges from 0.86 PgC yr⁻¹ for CCSM4 to 2.76 PgC yr⁻¹ for GFDL. When compared with observational products (Supplementary Figure S13h), we find that, excluding FLUXCOM, which is known to underestimate the global NBP IAV (see Jung et al. 2019, Biogeosciences), the CAMS atmospheric CO₂ inversion suggests a value of .68, while dynamic vegetation models used for the Global Carbon Project suggest a range of 0.53 to 1.50. Thus, some models (GFDL in particular) seem to overestimate the overall NBP variability. This is now discussed thoroughly in the Methods at lines L283-292.

However, all the models (including CCSM4 which is closest to the observational range) are unanimous that the global NBP variability is reduced by about 90% when prescribing soil moisture (Figure 1c) and that indirect effects dominate this response (Supplementary Table 2). It is in fact quite striking that, even though models may disagree in terms of their spatial patterns or global mean fluxes, they all agree that this is an important mechanism. Uncovering this mechanism at the global scale is only be possible with such climate model experiments (doing this with observations would require manipulating soil moisture everywhere on the planet in order to estimate the land-atmosphere feedback impacts).

Regarding the long-term average NBP, there is still a very large uncertainty with respect to the spatial patterns of net carbon fluxes, making it hard to make any relevant model evaluations. It certainly is a big issue, and this has been documented in several papers with both observational products or coupled climate model simulations, for instance in Chevallier et al. 2019, Atmospheric Chemistry and Physics, Jung et al. 2019, Biogeosciences, or Davies-Barnard et al. 2020, Biogeosciences. Reducing this uncertainty will likely remain a motivation for more scientific work over the next decades. Instead, we compared the simulated GPP against observational datasets, which are arguably somewhat better constrained for GPP than they are for NBP. This is presented in Supplementary Figure S21 (note that GPP is not available for the IPSL model). We find that the models agree very well with the observational data in terms of spatial patterns. For global mean GPP, two models produce a relatively high global GPP (about 150PgC yr⁻¹). Still, such values are not unrealistic according to other satellite-based estimates (e.g. Joiner et al. 2018 in Remote Sensing report 140PgC yr⁻¹). This is now discussed at lines L299-305.

6. L229: I think so this is a crucial part in the discussion. Indeed all models are parameterized and different parameterizations could might work better in coupled and uncoupled ways. But focus on the best estimate doesn't necessary help us in understanding the mechanisms. Besides the earlier mentioned debate on limitation versus stress function of course also the biome approach compared to the trade approach.

(note: L229 that is referred to by the reviewer is now L197). Thank you, we completely agree with this comment. Despite the fact that the models are imperfect representations of the Earth system, our results enable us to better understand a mechanism that apparently controls a substantial fraction of NBP IAV, regardless of the different implementations.

7. Figure 2: Why is it that the Congo basin is reacting differently compared to the other tropical forests? (Amazon, Asia)

Yes, this is an interesting question. We suppose that this occurs because simulated soil moisture has little inter-annual variability (relative to the seasonal cycle) in this region according to most of the models. Thus, prescribing seasonal soil moisture does not make a very big difference, leading to few changes in the carbon fluxes.

Referee #3 (Remarks to the Author):

The manuscript describes a modelling effort to ascertain how much variation in net carbon dioxide uptake by the terrestrial biosphere is associated with variation in soil moisture content directly, and how much is due to indirect effects via the associated reduction in evaporation, causing a drying and heating of the air passing across the vegetation. It is a useful contribution as unfortunately some scientists have ascribed effects in off-line models to, for example, an increase in warming, without recognizing that in the runs examined the warming was actually a response to reduced rainfall. This idea was well expressed earlier, and deserves quotation here, by Yin, D., Roderick, M.L., Leech, G., Sun, F. and Huang, Y., 2014. The contribution of reduction in evaporative cooling to higher surface air temperatures during drought. *Geophysical Research Letters*, 41(22): 2014GL062039.

Thank you for the positive evaluation of the manuscript and the useful suggestion. We have added this reference at line L54.

The present manuscript is careful to acknowledge that rainfall itself will be affected by changes in air flow patterns.

It is a little odd that soil moisture is separated from VPD and temperature, but the latter are not separated. Why not? Giving the responses to temperature, separated from VPD and soil moisture, would expose the modelled effect of temperature. This would be helpful, as it has tended to be lost in the complex of equations necessary to describe biosphere functioning.

Yes, we fully agree that separating the effects of VPD and temperature would have been interesting. Unfortunately, VPD and temperature anomalies are often well correlated with each other (more than they are with soil moisture anomalies) so that we have some reservations concerning the ability of the sensitivity analysis (Equation 6) to correctly separate the temperature and VPD contributions. This is the reason why these two contributions are analysed jointly in the paper. This is explained in greater detail at lines L410-416 in the Methods. Interested readers can assess the separated temperature and VPD contributions in Supplementary Figure S9. Disentangling temperature and VPD effects is something that we would have liked to explore, however, we believe it would be better achieved with a dedicated model experiment. We note that analysing temperature and VPD jointly still allows us to contrast the direct versus indirect effects of soil moisture variability, which is the main objective of the paper.

I realise that it is not my job to comment on the English, but there are a couple of places in the introduction where it is a little obscure.

Line 22 “changes in carbon uptake by terrestrial ecosystems play an essential role for atmospheric CO₂ concentrations.” I suggest: “changes in carbon uptake by terrestrial ecosystems play an essential role in determining atmospheric CO₂ concentrations”.

This suggestion is very much appreciated. We have corrected the wording at this location.

Line 31 explain “coupled’ when it first appears.

This term is now explained at L240-243.

Line 35 “is often studied by intervening solely on soil moisture content”. How about “is often studied by intervening solely IN soil moisture content”

Thank you, we have modified the sentence as suggested.

A more important point is the final conclusion at lines 37 and 38, which should read:” Our results suggest that most of the variability in MODELLED global land carbon uptake is driven by temperature and vapour pressure deficit effects which are controlled by soil moisture.” The ‘modelled’ is important as the observation that 60% of the effects come indirectly is totally sensitive to the modelling of vegetation responses. The experimental results of Wong et al. (Nature) had a big impact via the Ball-Berry equation linking stomatal conductance with relative humidity. The equations fitted well generally and together with slight modification by Leuning became very popular. There was nothing quantitatively equivalent in terms of the effects of soil moisture content. So the balance of impact has tended to be described via the easily measurable responses of gas exchange to VPD. My point is that the 60% value is a very soft, modelled one.

We agree with the reviewer on this point. While the sentence at L37-38 was removed to meet the abstract word count, this particular caveat is still clearly mentioned in the conclusion at line L190.

Line 43 Reference 4 is not really a theoretical advance. It is a convenient fix which assumes that leaves are always RuBP regeneration limited (when in fact in sunlight they are not), introduced to give the correct sense of response of stomatal conductance to changes in [CO₂].

Thank you for noting this, this reference is no longer cited as such.

Line 65. Susceptible to having

Thank you for the correction, the sentence has been modified.

I congratulate the authors.

Referee #4 (Remarks to the Author):

This paper uses results from coupled land-atmosphere simulations with a GCM ensemble to diagnose direct and indirect contributions of soil moisture to NBP variability of land ecosystems globally and regionally. By comparing a fully-coupled control simulation with a simulation in which an annual cycle of soil moisture is prescribed from climatology in each grid cell, the study infers that

interannual soil moisture variability is the dominant control on variation in NBP. Indirect effects mediated by effects of surface evaporation on near-surface temperature and humidity are found to be stronger than the direct effects of soil water availability on photosynthesis. Tropical and semi-arid biomes are identified as ‘hotspots’ for this driving pattern.

This study is a contribution to a running debate on the climatic controls of NBP variability in low latitude ecosystems, which in turn control interannual variations (IAV) in atmospheric anomalies of CO₂. A number of high-profile papers have been published, some asserting soil moisture control and others temperature control, the latter purportedly mediated by effects on respiration. The cited paper by Shilong Piao and coauthors (2019) reviewed most of the key papers and employed structural equation modelling to conclude that interactions between temperature and moisture availability, that is, the dependence of carbon cycle sensitivity to temperature on moisture conditions, was the dominant control of IAV. I think this is a realistic framing, which goes some way towards reconciling apparently conflicting findings of different studies as to the controlling drivers. In general studies that identify temperature as the key driver come to this conclusion based on a simple correlation of

NBP or a proxy such as RLS, IAV etc on large-scale climate (Anderegg 2015, Cox 2013). Bottom-up studies based on local flux observations or process modelling (Jung 2017, Ahlström 2015) tend to unearth more subtle causal relationships in which hydrological variations and large-scale circulation phenomena such as ENSO play an underpinning role.

The present paper adds to this existing tapestry of studies. Results are revealing but not unexpected given the known behavior of the models. The paper is the latest in a series based on the GLACE-CMIP5 ensemble, here comprising a subset of four CMIP5-generation GCMs. It should be noted that these models are now being superseded by a new generation being deployed for CMIP6, in many cases adopting more sophisticated schemes for land surface dynamics and biogeochemistry. That said, I believe the general behavior and driver sensitivity of the models will not have changed so much that the conclusions of this paper would change, were the analysis to be repeated using the latest ESM versions of the same models.

The paper has some similarity with Green et al (2019) also in Nature which used the GLACE-CMIP5 ensemble to highlight the role of soil moisture as a driver of NBP evolution in the context of an RCP8.5 21st century climate-emissions scenario. The present paper demonstrates high NBP sensitivity to soil moisture is upheld from multi-annual to decadal-centennial time scales. This will not necessarily be true of the real Earth system. Disregarding feedbacks to climate through the carbon cycle (not accounted for in either paper) there are still several reasons why we might expect shifts in the sensitivity to environmental drivers as we move up to longer time scales. A shift in NBP will affect the amount of respiring live biomass and dead organic matter over time. Shifts in vegetation demography and PFT distributions might affect phenology, hydraulics and other vegetation properties, in turn impacting productivity directly and via evapotranspiration feedback. The GLACE ensemble only incorporates the first item from this list (changing substrate for respiration), and it turns out this is not an important aspect at multi-annual scale, with the response of

GPP dominating the response of NBP (in line with e.g. the Piao et al analysis). While this paper and Green et al. form a useful companion set, conclusions are largely informed by the same environmental controls and biophysical feedback mechanisms playing out in the models. In my opinion, this is an issue for rating the novelty of the present paper.

I believe the inferential approach of the study, comparing a coupled and uncoupled model experiment (wrt soil moisture) is sound. However, I think the narrative claiming that the coupled land-atmosphere modelling reveals something new, not captured by other studies in this space, is overstated. The feedbacks that are explicit in a coupled model are implicit in the observations entering analyses such as Jung 2017, using upscaled flux tower data. In the case of offline modelling, the forcing data on temperature, rainfall etc incorporate real-world dependencies on soil moisture, evapotranspiration and energy balance. So long as the offline model has some skill in reproducing vegetation properties that control these quantities, and the quantities themselves, there is no reason to expect the influence of this coupling on the relationship between soil moisture and NBP would be missed. The main advantage of a coupled over an uncoupled approach is the ability to partition the overall relationship between a direct and coupled effect. I do not agree with the assertion on L170-173 that this quality of offline models to capture the full response will somehow break down when forced by GCM output fields in a future climate projection. The statement suggesting this refers to Figure 3c as supporting evidence, but here the coupling between soil moisture and air temperature and VPD was deliberately broken when the models were forced by a climatology of soil moisture from the control run. This will not be the case when output from a straight GCM projection (i.e. with physically consistent links between different output fields) is used as input to an offline model simulation.

Thank you very much for this detailed comment. We fully agree with the reviewer's argument about offline model runs. In fact, we do not argue that offline models miss any of the indirect effects. We realize that the presentation may have been misinterpreted in this respect. The reviewer is correct that offline models are typically forced with meteorological data that normally includes the dependencies of temperature and VPD on soil moisture implicitly.

The message we want to convey is that from an "offline" perspective, soil moisture will often appear to only be a minor driver of globally integrated NBP IAV (compared to temperature and VPD). It is easy to see how such a conclusion could be reached from observational estimates for example (see Supplementary Figure S12). Only the coupled analysis can show that a lot of the temperature and VPD effects are actually controlled by soil moisture variability. This is the message we wanted to convey. We have modified the figure labels of Figure 3c (now in Supplementary Figure S10) and made our statement at L170-173 of the original submission (now at L143-146) clearer in this respect: *"This finding reconciles opposing perspectives on the roles of temperature versus water availability, as the apparent importance of either driver actually depends on whether the indirect effects are attributed to temperature or soil moisture (Supplementary Figures S10-S11)"*.

In the context of climate change and the intense discussions in the community about the future of the land carbon sink, we believe this is an important element. Extremes in T and VPD are projected to increase faster in areas of strong land-atmosphere coupling. These faster increases are more driven by trends in precipitation and soil moisture rather than by the overall level of surface temperature warming. Our analysis shows that these indirect effects drive most of the net carbon uptake IAV.

The paper does make some good points about limitations of current land surface models that can be traced to their history (L230-237). Both coupled and offline models incorporate stomatal conductance parameterisations that capture empirical dependencies on temperature and VPD/humidity but there is no established standard for capturing sensitivity to soil moisture, with a range of approaches, ranging from simple scalars to complex, e.g. optimisation-based, approaches being deployed. A number of models substitute a direct dependence of rubisco capacity for which evidence is sparse, except under extreme conditions. However these criticisms apply equally to the vegetation schemes incorporated in coupled ESMs, which are usually a rewired version of an offline LSM or DGVM.

In summary, I find this a sound and well-presented paper that makes a useful contribution to an already fairly substantial body of global-scale studies seeking to identify the causal mechanisms behind biospheric control of NBP/IAV. Results appear to be consistent with Piao et al (2019) who arrived at the same conclusions using a combination of approaches, and also relates to the Green et al. (2019) study using the same model ensemble to address the longer-term sensitivity of NBP to its candidate drivers. The cited references make appropriate credit to previous work.

Thank you very much for the clear and substantiated review.

Reviewer Reports on the First Revision:

Ref #1

I am happy with the revisions. I find the paper provides an elegant analysis and novel insights into the principal causes of interannual variability. While we knew that soil moisture was a major cause of IAV, and that variations in soil moisture can drive extremes in T and VPD, the study demonstrates that the feedback of variations in soil moisture to T and VPD is quantitatively important for NBP, which I believe to be a new insight into the functioning of the Earth system. As the authors state, there are a number of implications for the design and interpretation of experimental and observational studies into vegetation responses to extremes.

I have just a couple more suggestions to improve the presentation.

Line 70: It would be useful to give a very brief indication as to what the GLACE-CMIP5 experiment

was about here in the main text. It is not explained in the methods either. A sentence or two here would help the reader to understand what has been done.

Fig 1b: I was hoping to see a similar analysis for T, VPD and Rnet. Timecourses are shown in Fig S17, but I would like to see the summary SD for these environmental drivers. In particular, it looks as if the change in SD between CTRL and ExpA is larger for VPD than for T. The authors note that it is difficult to separate T and VPD given their strong correlation, but there is also great interest at the moment among experimental plant scientists in the relative roles of T and VPD, especially given that this strong correlation may shift over time. Some new evidence recently came to light to indicate that VPD may be a stronger driver than T of GPP at high temperatures (Smith et al. <https://www.nature.com/articles/s41477-020-00780-2>)

Hence, it would be good to see the relative changes in SD of T and VPD, and it might be nice to add a comment to the discussion about their relative importance and the potential for the T-VPD relationship to change over time.

Ref #2

Although the authors gave clear answers to all questions raised by the three reviewers, the most important issue that the conclusions are only based on models is only scratched at the surface. The authors agree that: "the model is an imperfect representation and now show in the best way comparisons with real data". But didn't ask the fundamental question if you can derive such strong conclusions from imperfect models?

I fully agree with the authors why this research is so important, as they cite "Because soil and atmospheric dryness do not equally respond to climate change, disentangling these direct and indirect effects seems a useful contribution and improves our overall understanding of what mechanisms drive the year to- year variability of land carbon uptake"

This is exactly what should be done and should be tested with observations, for instance in a similar system with a period of soil dryness and a period with atmospheric dryness.

The authors conclude that in particular, the result that direct soil moisture impacts are minor relative to the impacts of atmospheric feedbacks to temperature and VPD is new, although they ignore their own findings in here from Green et al. 2020 with overlap in co-authors, who for instance conclude: "Our results show that land surface models used for climate projections are overestimating atmospheric water stress in the tropical rainforests due in large part to the absence of dynamic vegetation biogeochemistry"

I disagree with the findings of the authors as the conclusions is only based on model results which are imperfect. There is no prove in the paper, based on a review of findings from observations, that their claim is real. We are just not certain that it is the case in reality and thereby not breakthrough research.

Is it that these model results are not valuable at all? No, of course the model results are highly usefull, as with the mass balance equations the effects of soil moisture and VPD can be used to mimick those in closed greenhouse settings with detailed plant physiological analyses for instance. An important next step should be to combine the findings from models with those of observational evidence. Green et al (2020) show that dynamical changes in canopy structure can maybe entirely compensate the dryness response of VPD on GPP from a leaf-level understanding. On the other hand, discussion on bias in RS products on GPP and droughts are also becoming more and more important, for instance Stocker et al. (2019, Nature Geoscience). And the FACE data analyses also have shown that parameterizations of the used ESM models are potentially wrong (Fatichi et al. PNAS 2016). having impact on the parameterization of the ESM models.

To conclude it is good work, but not novel and breakthrough research. The steps to go is go back

to the fundamental mechanisms in the models and find evidence from different disciplines including plant physiology to understand how vegetation reacts. Their conclusions are not robust

Ref #3

The authors have not attempted to separate the effects of VPD and T. This is disappointing. It would be entirely within the flavour of the present manuscript, but more complete, with that addition. My other comments on the first version were trivial and have been addressed.

Ref #4

I enjoyed reading the updated version of this interesting manuscript. As pointed out in my earlier review, this is a well-crafted paper that adds to the existing literature trying to pin down the climatic controls of global/tropical NBP variation and associated variation in atmospheric CO₂. This study adds to understanding of the role of soil moisture as an underlying control on vegetation responses. It is pitched as reconciling 'conflicting evidence on the roles of temperature versus water availability' in controlling NBP variation at global scale. This is achieved through a model experiment in which evapotranspiration-mediated coupling of soil moisture status to atmospheric temperature and humidity is switched on and off in alternate simulations which are then compared. Results are consistent with the findings of a review by Piao et al. (2019) and closely related to Green et al. (2019) which looked at responses to an RCP8.5 scenario using the same model ensemble.

In my view, this paper constitutes a useful addition to the existing body of literature linking NBP to hydrological variations. However, it is really only connecting the dots between a number of existing studies using the same (Green) or complementary (Piao) approaches. Results are not unexpected given these earlier papers: the findings consolidate rather than advance knowledge. In this light it could be questioned whether publication in Nature is motivated.

The authors have improved clarity on one of the points I raised in my earlier review, namely the degree to which the LSM-based approach captures land-atmosphere feedbacks (i.e. indirect effects) not accounted for by observation-based and offline vegetation model studies. I am happy that authors have clarified that their approach comparing two model experiments serves to disentangle the direct from the indirect effects, recognising that the indirect effects are included (implicit) in the data using other methods, both observational and offline-model based.

A further specific point from my earlier review, which the authors do not acknowledge in their rebuttal, concerned the scope of the indirect effects or feedbacks captured by the GLACE ensemble. I was perhaps insufficiently clear, but was hoping some discussion would be added to the paper acknowledging that, while short-term biophysical feedbacks are captured through the LSMs, longer-term feedbacks affecting vegetation structure/composition/phenology and soil water retention properties - which in turn affect hydrological cycling - are not. These longer-term feedbacks undoubtedly have some influence on flux tower and atmospheric concentration data used in observational studies of NBP variation, and would certainly be relevant for the centennial timescale of the Green paper. Some mention of these issues would strengthen the linkage of this paper to the wider literature, as well as pointing out a direction for useful future work using the next generation of coupled ESMs that accommodate a wider range of feedbacks. A suitable point to add mention of the scope of feedbacks considered would be around the third paragraph (beginning "Soil moisture drought ..."), as well as in connection to the interpretation of results in terms of direct and indirect effects.

Author Rebuttals to First Revision:

This document contains:

3. Overview of minor changes
4. Detailed response to referees' comments and suggestions

3. Overview of minor changes

4. In response to suggestions by the editor, the title of the paper was modified to: "Soil moisture–atmosphere feedbacks dominate land carbon uptake variability". The summary paragraph was also altered to improve wording and clarity.
5. In response to referee #1, we have added an analysis equivalent to Fig 1b in Supplementary Fig. 8. We also discuss the limitations of the modelling approach more extensively in the main text at lines L169-175 in response to referee #2, and have included a discussion of long-term feedbacks as suggested by referee #4 at L183-185.
6. Some content in the Methods has been reorganized for clarity. For instance, a separate section on the "Joint analysis of T and VPD effects" now includes a discussion on the individual contributions of T and VPD as suggested by referees #1 and #3.
7. Panel d of Figure 2 was moved back to Extended Data Fig. 6. This was done because this figure panel was only briefly referred to in the main text and is not central to the main findings.

4. Detailed response to referees' comments and suggestions

Referee #1 (Remarks to the Author):

I am happy with the revisions. I find the paper provides an elegant analysis and novel insights into the principal causes of interannual variability. While we knew that soil moisture was a major cause of IAV, and that variations in soil moisture can drive extremes in T and VPD, the study demonstrates that the feedback of variations in soil moisture to T and VPD is quantitatively important for NBP, which I believe to be a new insight into the functioning of the Earth system. As the authors state, there are a number of implications for the design and interpretation of experimental and observational studies into vegetation responses to extremes.

I have just a couple more suggestions to improve the presentation.

Line 70: It would be useful to give a very brief indication as to what the GLACE-CMIP5 experiment was about here in the main text. It is not explained in the methods either. A sentence or two here would help the reader to understand what has been done.

Thank you for the positive review. We have added an explanation about the original purpose of the GLACE-CMIP5 experiments in the Methods at L353-356: *“This model experiment was originally designed to investigate soil moisture – climate feedbacks under historical and future scenarios, and notably their impact on extreme heat events⁶. Its experimental design is inspired from the original GLACE experiment⁴³, which focused on the role of soil moisture in seasonal weather predictability.”*

Fig 1b: I was hoping to see a similar analysis for T, VPD and Rnet. Time courses are shown in Fig S17, but I would like to see the summary SD for these environmental drivers. In particular, it looks as if the change in SD between CTRL and ExpA is larger for VPD than for T.

Thank you very much for these comments. We now show the summary change in SD between CTL and ExpA for global mean T, VPD and Rnet as well in Fig S17 (now Supplementary Fig. 8).

The authors note that it is difficult to separate T and VPD given their strong correlation, but there is also great interest at the moment among experimental plant scientists in the relative roles of T and VPD, especially given that this strong correlation may shift over time. Some new evidence recently came to light to indicate that VPD may be a stronger driver than T of GPP at high temperatures (Smith et al. <https://www.nature.com/articles/s41477-020-00780-2>). Hence, it would be good to see the relative changes in SD of T and VPD, and it might be nice to add a comment to the discussion about their relative importance and the potential for the T-VPD relationship to change over time.

Thank you very much for this comment. As acknowledged by the referee, separating the contributions of T and VPD to NBP is not straightforward within the study’s design. However, given the interest (also mentioned by referee #3) of discussing the relative importance of T and VPD (which was already presented in Extended Data Fig. 4-5 but was not discussed), we now mention this aspect in the main text at L113-115 and provide a brief discussion in the Methods at L546-557: *“With the caveats mentioned above, Extended Data Fig. 4 shows that VPD has a much larger role than T in the reduction of NBP variability occurring between CTL and ExpA. However, this does not mean that T is less sensitive than VPD to prescribing soil moisture. Rather, Extended Data Fig. 5 shows that the sensitivity analysis attributes more NBP variability to VPD to begin with but that both the VPD-driven and T-driven NBP variability are reduced in ExpA.”*

Referee #2 (Remarks to the Author):

Dear editor,

Although the authors gave clear answers to all questions raised by the three reviewers, the most important issue that the conclusions are only based on models is only scratched at the surface. The authors agree that: “the model is an imperfect representation and now show in the best way comparisons with real data”. But didn’t ask the fundamental question if you can derive such strong conclusions from

imperfect models?

I fully agree with the authors why this research is so important, as they cite “Because soil and atmospheric dryness do not equally respond to climate change, disentangling these direct and indirect effects seems a useful contribution and improves our overall understanding of what mechanisms drive the year-to-year variability of land carbon uptake”

This is exactly what should be done and should be tested with observations, for instance in a similar system with a period of soil dryness and a period with atmospheric dryness.

Thank you for your comments. We appreciate these concerns. We would like to stress that the mechanisms and feedbacks evidenced in our study are by far not just based on model results but also supported by numerous observational case-studies. For instance, the role of soil moisture drought in generating warm and dry atmospheric conditions has been documented using both in-situ weather station measurements, satellite data, and global bias-corrected weather datasets (e.g. Hirschi et al. 2011, Miralles et al. 2014, Dirmeyer 2011). Similarly, the fact that ecosystems and carbon exchange can be sensitive to direct water stress effects, high temperatures, and vapour pressure deficits is very well established (e.g. Novick et al. 2016, Rogers et al. 2017).

Thus the main mechanisms and processes which our results rely upon are supported by observational evidence. The model experiment presented here serves to investigate the global relative importance of these physical mechanisms. Such factorial Earth system model simulations represent a well-established scientific approach and, often, they are also our only opportunity of testing the globally integrated impact of certain variables or physical processes in the Earth system, providing insights into key scientific questions with broad relevance (e.g. IPCC 2013, Arora et al. 2020).

As mentioned by the referee, we fully acknowledge and clearly state in the paper that our results are based on Earth system model simulations, now also in the abstract (e.g. at L34, L160-162) and that these simulations are imperfect representations (now discussed more extensively at L169-175). To the extent possible, we have also evaluated these simulations against various (and also imperfect) global observational estimates (L131-138, L406-436, Extended Data Fig. 8, Supplementary Figs. 6, and 9-12).

Hirschi, M., Seneviratne, S. I., Alexandrov, V., Boberg, F., Boroneant, C., Christensen, O. B., ... & Stepanek, P. (2011). Observational evidence for soil-moisture impact on hot extremes in southeastern Europe. *Nature Geoscience*, 4(1), 17-21.

Miralles, D. G., Teuling, A. J., Van Heerwaarden, C. C., & De Arellano, J. V. G. (2014). Mega-heatwave temperatures due to combined soil desiccation and atmospheric heat accumulation. *Nature geoscience*, 7(5), 345-349.

Dirmeyer, P. A. (2011). The terrestrial segment of soil moisture–climate coupling. *Geophysical Research Letters*, 38(16).

Novick, K. A., Ficklin, D. L., Stoy, P. C., Williams, C. A., Bohrer, G., Oishi, A. C., ... & Scott, R. L. (2016). The increasing importance of atmospheric demand for ecosystem water and carbon fluxes. *Nature climate change*, 6(11), 1023-1027.

Rogers, A., Medlyn, B. E., Dukes, J. S., Bonan, G., Von Caemmerer, S., Dietze, M. C., ... & Prentice, I. C. (2017). A roadmap for improving the representation of photosynthesis in Earth system models. *New Phytologist*, 213(1), 22-42.

IPCC, 2013: *Climate Change 2013: The Physical Science Basis. Contribution of Working Group I to the Fifth Assessment Report of the Intergovernmental Panel on Climate Change* [Stocker, T.F., D. Qin, G.-K. Plattner, M. Tignor, S.K. Allen, J. Boschung, A. Nauels, Y. Xia, V. Bex and P.M. Midgley (eds.)]. Cambridge University Press, Cambridge, United Kingdom and New York, NY, USA, 1535 pp, doi:10.1017/CBO9781107415324.

Arora, V. K., Katavouta, A., Williams, R. G., Jones, C. D., Brovkin, V., Friedlingstein, P., ... & Chamberlain, M. A. (2020). Carbon-concentration and carbon-climate feedbacks in CMIP6 models and their comparison to CMIP5 models. *Biogeosciences*, 17(16), 4173-4222.

The authors conclude that in particular, the result that direct soil moisture impacts are minor relative to the impacts of atmospheric feedbacks to temperature and VPD is new, although they ignore their own findings in here from Green et al. 2020 with overlap in co-authors, who for instance conclude: “Our results show that land surface models used for climate projections are overestimating atmospheric water stress in the tropical rainforests due in large part to the absence of dynamic vegetation biogeochemistry”

We do not think that the findings in Green et al. 2020 contradict the results shown here. We note that this statement from Green et al. 2020 refers to a positive SIF response to VPD observed during the wet season over the Amazon rainforest. However, it provides no evidence that this type of response affects NBP variability worldwide or that it would constitute a dominant driver of global inter-annual variability in net carbon exchange. Also, Green et al. 2020 do not disentangle the relative contributions of direct versus indirect (feedback) effects, which is a main novelty of this study compared to previous work.

While Green et al. 2020 do highlight that some processes are still missing in Earth system models and might explain the peculiar SIF response over the Amazon, they also show for instance that both observed SIF and modelled GPP agree on a negative response to VPD in the neighbouring regions of South America. They also report that modelling a stomatal physiological stress in response to higher VPD (as done in current models) is consistent with the flux tower observations.

I disagree with the findings of the authors as the conclusions is only based on model results which are imperfect. There is no prove in the paper, based on a review of findings from observations, that their claim is real. We are just not certain that it is the case in reality and thereby not breakthrough research.

See the first part of our response above and the references therein which provide observational evidence for the main mechanisms and processes discussed in the study.

Is it that these model results are not valuable at all? No, of course the model results are highly useful, as with the mass balance equations the effects of soil moisture and VPD can be used to mimic those in closed greenhouse settings with detailed plant physiological analyses for instance.

An important next step should be to combine the findings from models with those of observational evidence. Green et al (2020) show that dynamical changes in canopy structure can maybe entirely compensate the dryness response of VPD on GPP from a leaf-level understanding. On the other hand, discussion on bias in RS products on GPP and droughts are also becoming more and more important, for instance Stocker et al. (2019, Nature Geoscience). And the FACE data analyses also have shown that parameterizations of the used ESM models are potentially wrong (Fatichi et al. PNAS 2016). having impact on the parameterization of the ESM models.

Thank you for this feedback. We agree that these are interesting and crucial challenges for the next generation of Earth System models and observing systems. We believe that because our work demonstrates the global importance of soil moisture – atmosphere feedbacks for NBP IAV in current models, it will also motivate and justify future work in such directions. We also agree with the reviewer that, in the long term, vegetation can adapt to changes in environmental conditions, for instance to rising atmospheric CO₂ concentrations, with consequences for soil moisture – atmosphere interactions. This is now discussed in the main text at lines L183-185: *“We also note that long-term changes in vegetation structure and composition might alter the ecosystem’s future response⁴ to and control^{9,48,49} of soil moisture–atmosphere feedbacks.”*

To conclude it is good work, but not novel and breakthrough research. The steps to go is go back to the fundamental mechanisms in the models and find evidence from different disciplines including plant physiology to understand how vegetation reacts. Their conclusions are not robust.

Referee #3 (Remarks to the Author):

The authors have not attempted to separate the effects of VPD and T. This is disappointing. It would be entirely within the flavour of the present manuscript, but more complete, with that addition. My other comments on the first version were trivial and have been addressed.

Thank you for this feedback. Given the interest (also mentioned by referee #1) of discussing the relative importance of T and VPD (which was already presented in Extended Data Fig. 4-5 but was not discussed), we now mention this aspect in the main text at L113-115 and provide a brief discussion in the Methods at L546-557: *“With the caveats mentioned above, Extended Data Fig. 4 shows that VPD has a much larger role than T in the reduction of NBP variability occurring between CTL and ExpA. However, this does not mean that T is less sensitive than VPD to prescribing soil moisture. Rather, Extended Data Fig. 5 shows that the sensitivity analysis*

attributes more NBP variability to VPD to begin with but that both the VPD-driven and T-driven NBP variability are reduced in ExpA.”

Referee #4 (Remarks to the Author):

I enjoyed reading the updated version of this interesting manuscript. As pointed out in my earlier review, this is a well-crafted paper that adds to the existing literature trying to pin down the climatic controls of global/tropical NBP variation and associated variation in atmospheric CO₂. This study adds to understanding of the role of soil moisture as an underlying control on vegetation responses. It is pitched as reconciling 'conflicting evidence on the roles of temperature versus water availability' in controlling NBP variation at global scale. This is achieved through a model experiment in which evapotranspiration-mediated coupling of soil moisture status to atmospheric temperature and humidity is switched on and off in alternate simulations which are then compared. Results are consistent with the findings of a review by Piao et al. (2019) and closely related to Green et al. (2019) which looked at responses to an RCP8.5 scenario using the same model ensemble.

In my view, this paper constitutes a useful addition to the existing body of literature linking NBP to hydrological variations. However, it is really only connecting the dots between a number of existing studies using the same (Green) or complementary (Piao) approaches. Results are not unexpected given these earlier papers: the findings consolidate rather than advance knowledge. In this light it could be questioned whether publication in Nature is motivated.

Thank you for the positive feedback. While Green et al. showed the strong role of soil moisture for long-term NBP, it did not disentangle the direct effects of water stress versus the indirect response to the atmospheric feedback. We show here that making this distinction is central to reconciling these previous assessments of the relative roles of soil moisture versus temperature/VPD. The importance of the indirect effects shown here cannot be inferred from the results of Green et al. 2019 or the review by Piao et al. 2019 in our opinion. This is now more clearly stated at L167-169: “*Our results reveal that soil moisture–atmosphere feedbacks represent a dominant source of variability in global carbon uptake and thus reconcile previous conflicting assessments²⁻⁵”*. We believe that this improvement in our general understanding of the potential causal pathways (e.g. Figure 3a) will also be beneficial to the interpretation and analysis of many types of regional and in-situ observations of carbon exchange and vegetation status.

Finally, our results are also not entirely in line with those of Piao et al., who argue based on empirical analyses that the sensitivity of carbon fluxes to temperature is dependent on the soil moisture status (i.e. $NBP \sim \beta * T$, with $\beta \sim f(SM)$). This type of relationship is also hypothesized for instance in Wang et al. 2014, Nature. On the contrary, our results rather hint that it is the temperature and VPD variability itself that is controlled by soil moisture through the atmospheric feedback (i.e. $T \sim f(SM, \dots)$). Thus even though both studies agree that soil moisture plays an important role for carbon exchange variability, the proposed causal mechanism is actually quite different, with important implications for studies using this sensitivity as an

observational constraint in our opinion. This is now more clearly explained at L148-158.

The authors have improved clarity on one of the points I raised in my earlier review, namely the degree to which the LSM-based approach captures land-atmosphere feedbacks (i.e. indirect effects) not accounted for by observation-based and offline vegetation model studies. I am happy that authors have clarified that their approach comparing two model experiments serves to disentangle the direct from the indirect effects, recognising that the indirect effects are included (implicit) in the data using other methods, both observational and offline-model based.

A further specific point from my earlier review, which the authors do not acknowledge in their rebuttal, concerned the scope of the indirect effects or feedbacks captured by the GLACE ensemble. I was perhaps insufficiently clear, but was hoping some discussion would be added to the paper acknowledging that, while short-term biophysical feedbacks are captured through the LSMs, longer-term feedbacks affecting vegetation structure/composition/phenology and soil water retention properties - which in turn affect hydrological cycling - are not. These longer-term feedbacks undoubtedly have some influence on flux tower and atmospheric concentration data used in observational studies of NBP variation, and would certainly be relevant for the centennial timescale of the Green paper. Some mention of these issues would strengthen the linkage of this paper to the wider literature, as well as pointing out a direction for useful future work using the next generation of coupled ESMs that accommodate a wider range of feedbacks. A suitable point to add mention of the scope of feedbacks considered would be around the third paragraph (beginning "Soil moisture drought ..."), as well as in connection to the interpretation of results in terms of direct and indirect effects.

Thank you for elaborating on this previous point. We fully agree that such long-term feedbacks will play an important role especially when looking at long-term (centennial) changes in NBP and its sensitivity to climate variability. We now mention this in the main text at L183-185: "*We also note that long-term changes in vegetation structure and composition might alter the ecosystem's future response⁴ to and control^{9,48,49} of soil moisture-atmosphere feedbacks.*"